# Watermark-based Attribution of AI-Generated Content

**Zhengyuan Jiang, Moyang Guo, Yuepeng Hu, Yupu Wang, Neil Zhenqiang Gong**
Duke University
{zhengyuan.jiang,moyang.guo,yuepeng.hu,yupu.wang,neil.gong}@duke.edu

## Abstract

Several companies have deployed watermark-based detection to identify AI-generated content. However, attribution–the ability to trace back to the user of a generative AI (GenAI) service who created the given AI-generated content–remains largely unexplored despite its growing importance. In this work, we aim to bridge this gap by conducting the first systematic study on watermark-based, user-level attribution of AI-generated content. Our key idea is to assign a unique watermark to each user of the GenAI service and embed this watermark into the AI-generated content created by that user. Attribution is then performed by identifying the user whose watermark best matches the one extracted from the given content. This approach, however, faces a key challenge: How should watermarks be selected for users to maximize attribution performance? To address the challenge, we first theoretically derive lower bounds on detection and attribution performance through rigorous probabilistic analysis for any given set of user watermarks. Then, we select watermarks for users to maximize these lower bounds, thereby optimizing detection and attribution performance. Our theoretical and empirical results show that watermark-based attribution inherits both the accuracy and (non-)robustness properties of the underlying watermark. Specifically, attribution remains highly accurate when the watermarked AI-generated content is either not post-processed or subjected to common post-processing such as JPEG compression, as well as black-box adversarial post-processing with limited query budgets.

## 1 Introduction

Generative AI (GenAI)–such as DALL-E, Midjourney, and ChatGPT–can produce highly realistic content in various forms, such as images, text, and audio. While GenAI offers numerous societal benefits, it also raises significant ethical concerns. For example, it can be misused to create harmful content, support disinformation and propaganda campaigns by generating realistic-looking fake material (Dhaliwal, 2023), and enable individuals to falsely claim copyright ownership of AI-generated content (Escalante-De Mattei, 2023).

Watermark-based detection of AI-generated content has emerged as a promising technique to proactively address these ethical concerns. Specifically, a watermark is embedded into all content generated by a GenAI service, and the content is identified as AI-generated if a similar watermark can be decoded from it. Due to its potential, watermark-based detection has garnered significant attention. Several major companies have already deployed watermark-based detection, including OpenAI's DALL-E (Ramesh et al., 2022), Google's SynthID (Gowal & Kohli, 2023), Microsoft's Bing (Mehdi, 2023), and Stable Diffusion (Rombach et al., 2022). However, the generated content cannot be traced in further details, as the same watermark is applied to all generated content.

*Attribution* seeks to trace the origin of an AI-generated content, specifically identifying the user of the GenAI service who created it.[1] This capability is increasingly important in real-world deployments, where GenAI services face substantial misuse risks: malicious users can generate political deepfakes, deceptive propaganda, or other harmful content and distribute it anonymously. In

---

[1]Attribution can also refer to identifying the GenAI service responsible for generating the content, as discussed in Appendix O.

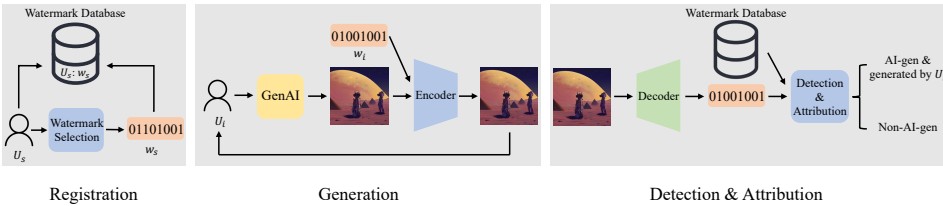

Figure 1: *Registration*, *generation*, and *detection & attribution* phases.

such scenarios, detection alone is insufficient—knowing that the content is AI-generated does not reveal *who* produced it. Attribution provides service providers and law-enforcement agencies with actionable forensic signals that link harmful content back to the originating user. This enables concrete interventions, such as identifying responsible users during abuse investigations, suspending or banning repeat offenders, and emerging regulatory requirements on content traceability. Thus, attribution complements detection: detection identifies *whether* the content is AI-generated, while attribution identifies *who* created it. Both are essential for building practical, enforceable, and responsible GenAI ecosystems. Despite the increasing importance, attribution remains largely unexplored.

**Our work:** In this work, we bridge this gap by conducting the *first* systematic study on watermark-based, user-level attribution of AI-generated content. Figure 1 illustrates our core idea. When a user registers with a GenAI service, the service provider assigns the user a unique watermark (i.e., a bitstring) and stores it in a watermark database. Whenever the user generates content using the GenAI service, their specific watermark is embedded in the content. The content is attributed to the user whose watermark is the most similar to the one extracted from the content. Our approach differs from standard user-agnostic watermark-based detection, where the same watermark is embedded in all AI-generated content. Thus, our approach also introduces a slight modification in the detection process: The content is identified as generated by the GenAI service if the extracted watermark closely matches *at least* one user's watermark.

However, watermark-based attribution faces a significant challenge: how should watermarks be selected for users to maximize detection and attribution performance? A straightforward solution is to empirically evaluate the detection and attribution performance for different numbers of users and corresponding watermark sets. However, this method is difficult to scale due to the vast space of possible watermarks. For instance, a 64-bit watermark yields $2^{64}$ potential watermarks, resulting in $\binom{2^{64}}{s}$ possible watermark sets for $s$ users. Evaluating the detection and attribution performance for each watermark set to find the optimal one is computationally infeasible.

We propose a two-step solution to address this challenge. First, we conduct a rigorous probabilistic analysis to theoretically evaluate user-aware detection and attribution performance for *any* given set of user watermarks. Specifically, we formally quantify the behavior of watermarking and derive lower bounds for the *true detection rate (TDR)* and *true attribution rate (TAR)*, as well as an upper bound for the *false detection rate (FDR)*. We also show that other relevant performance metrics can be derived from these three core metrics.

In the second step of our solution, we select watermarks for users to maximize the lower bounds of TDR and TAR, thereby optimizing TDR and TAR. Based on the analytical forms of these lower bounds, this optimization simplifies to selecting the most dissimilar watermarks for users. Formally, we define the *watermark selection problem*, which seeks to assign a watermark to a new user by minimizing the maximum similarity between the new watermark and those of existing users. Specifically, we adapt the *bounded search tree algorithm* (Gramm et al., 2003), an albeit inefficient and exact solution for the well-known *farthest string problem* (Lanctot et al., 2003), into an efficient approximate algorithm for watermark selection. We empirically evaluate our method on non-AI-generated images from three standard image benchmark datasets and AI-generated images produced by three GenAI models: Stable Diffusion, Midjourney, and DALL-E. For watermarking methods, we use HiDDeN (Zhu et al., 2018), StegaStamp (Tancik et al., 2020), and PRC (Gunn et al., 2025). Our results show that detection and attribution are highly accurate, with TDR and TAR close to 1 and FDR near 0, when AI-generated images are not post-processed, even for GenAI services with a large user base (e.g., 100 million users). Performance remains high under post-processing operations such as JPEG compression. Additionally, adversarial post-processing (Jiang et al., 2023b) with a limited number of queries to the detection API significantly degrades image quality to evade detection and attribution. Furthermore, our watermark selection algorithm outperforms baseline methods.

## 2 RELATED WORK

**Image watermarks:** An image watermarking method typically consists of three components: the *watermark*, the *encoder*, and the *decoder*. The watermark is usually represented as a bit string. An encoder $E$ embeds a watermark $w$ into an image, while a decoder $D$ attempts to recover the watermark from a (potentially watermarked) image. If the image is embedded with watermark $w$, the decoded output should closely match $w$. Some approaches design the encoder and decoder based on hand-crafted heuristics (Pereira & Pun, 2000; Bi et al., 2007; Wang, 2021). For instance, TreeRing (Wen et al., 2023) embeds a fixed pattern into the initial noise vector used in diffusion-based image generation and later extracts it for detection. In its original design, TreeRing applies a predefined Fourier-space pattern, resulting in a single-bit watermark. Subsequent studies (Yang et al., 2024b; Gunn et al., 2025) have extended this idea to embed multiple bits. For example, the PRC watermark (Gunn et al., 2025) samples the initial noise vector using a pseudorandom error-correcting code to represent multiple bits, which can then be recovered through the inverse diffusion process. In contrast to these methods, some (Kandi et al., 2017; Zhu et al., 2018; Wen & Aydore, 2019; Luo et al., 2020; Abdelnabi & Fritz, 2021; Fernandez et al., 2022; Jiang et al., 2023a; Lukas & Kerschbaum, 2023; Rezaei et al., 2024; Guo et al., 2024) use neural networks as the encoder/decoder and automatically learn them using an image dataset. For instance, HiDDeN (Zhu et al., 2018) jointly trains CNN-based encoder and decoder networks to embed and recover watermark bit strings. StegaStamp (Tancik et al., 2020) leverages deep neural networks to learn an encoding/decoding algorithm that is particularly robust against real-world post-processing, such as printing and photography. Stable Signature (Fernandez et al., 2023) integrates the encoder and watermark directly into the parameters of a diffusion model, ensuring that its generated images are inherently watermarked. Our theoretical analysis and watermark selection algorithm can be applied to any multi-bit watermarking method.

**Robustness of image watermarks:** We stress that our work builds on an existing watermarking method for user-level attribution of AI-generated images. Developing new watermarking techniques that are robust against post-processing–i.e., enabling watermark extraction even after the watermarked image has been modified–is orthogonal to our work. Notably, while creating fully robust watermarks remains an ongoing challenge, the field has made significant progress in recent years. For example, recent advancements in image watermarks (Tancik et al., 2020; Fernandez et al., 2023; Gunn et al., 2025; Lu et al., 2025; Sander et al., 2025; Ci et al., 2025) have shown robust performance against common post-processing. Notably, some watermarks (Jiang et al., 2024) are even certifiably robust, ensuring resilience against any post-processing that introduces bounded perturbations to the images. These image watermarking methods are not yet robust to adversarial post-processing in the white-box setting (Jiang et al., 2023b; Lukas et al., 2024; Hu et al., 2024) where an attacker has access to the decoder. However, they have good robustness to adversarial post-processing when an attacker can only query the detection API for a small number of times in the black-box setting (Jiang et al., 2024) or the attacker has limited computation resource to train a small number of surrogate models in transfer attacks (Hu et al., 2025). In particular, adversarial post-processing substantially decreases the quality of a watermarked image in order to remove the watermark in such scenarios.

## 3 PROBLEM FORMULATION

Suppose a GenAI model is deployed as a GenAI cloud service. A registered user sends a *prompt* (i.e., a text) to the GenAI service, which returns an AI-generated content to the user. *Detection* of AI-generated content aims to decide whether the given content was generated by the GenAI service or not; while *attribution* further traces back the user of the GenAI service who generated the content detected as AI-generated. Such attribution can aid the GenAI service provider or law enforcement in forensic investigations of cyber-crimes, e.g., disinformation or propaganda campaigns, that involve AI-generated content. We formally define the detection and attribution problems as follows:

**Definition 1** (Detection of AI-generated content)**.** Given the content and a GenAI service, detection aims to infer whether the content was generated by the GenAI service or not.

**Definition 2** (Attribution of AI-generated content)**.** Given the content, a GenAI service, and $s$ users $U = \{U_1, U_2, \cdots, U_s\}$ of the GenAI service, attribution aims to further infer which user used the GenAI service to generate the content after it is detected as AI-generated.

We note that the set of $s$ users $U$ in attribution could include all registered users of the GenAI service, in which $s$ may be very large. Alternatively, this set may consist of a smaller number of registered users if the GenAI service provider has some prior knowledge on its registered users. For instance, the GenAI service provider may exclude the registered users, who are verified offline as trusted, from the set $U$ to reduce its size. Moreover, malicious users may be identified by conventional network security solutions, such as IP addresses and behavior patterns (Stringhini et al., 2015; Yuan et al., 2019; Xu et al., 2021). How to construct the set of users $U$ in attribution is out of the scope of this work. Given any set $U$, our method aims to infer which user in $U$ may have generated the given content. We also note that another relevant attribution problem is to trace back the GenAI service that generated the given content. Our method can also be used for such GenAI-service attribution, which we discuss in Appendix O.

## 4 Watermark-based Attribution

Figure 1 illustrates our watermark-based detection and attribution of AI-generated content. When a user registers in the GenAI service, the service provider selects a unique watermark for the user. We denote by $w_i$ the watermark selected for user $U_i$, where $i$ is the user index. During content generation, when a user $U_i$ sends a prompt to the GenAI service to generate the content, the provider uses the watermark encoder $E$ to embed watermark $w_i$ into the content. During detection and attribution, a watermark is decoded from the given content; the given content is identified as generated by the GenAI service if the extracted watermark closely matches at least one user's watermark; and the given content is further attributed to the user whose watermark is the most similar to the decoded watermark after it is detected as AI-generated.

**Detection:** We use *bitwise accuracy* to measure similarity between two watermarks. Specifically, given any two watermarks $w$ and $w'$, their bitwise accuracy (denoted as $BA(w, w')$) is the fraction of matched bits in them: $BA(w, w') = \frac{1}{n} \sum_{k=1}^{n} \mathbb{I}(w[k] = w'[k])$, where $n$ is the watermark length, $w[k]$ is the $k$-th bit of $w$, and $\mathbb{I}$ is the indicator function. Given the content $C$, we use the decoder $D$ to decode a watermark $D(C)$ from it. We detect $C$ as AI-generated if there exists a user's watermark that is similar enough to $D(C)$, i.e., if the following is satisfied: $\max_{i \in \{1,2,\cdots,s\}} BA(D(C), w_i) \geq \tau$, where $\tau > 0.5$ is the *detection threshold*.

**Attribution:** Attribution is applied only after the content $C$ is detected as AI-generated. Intuitively, we attribute the content to the user whose watermark is the most similar to the decoded watermark $D(C)$. Formally, we attribute $C$ to user $U_{i^*}$, where $i^*$ is as follows: $i^* = \arg\max_{i \in \{1,2,\cdots,s\}} BA(D(C), w_i)$.

**Key challenge:** Watermark-based attribution faces a key challenge: how to select watermarks for users to maximize detection and attribution performance. To tackle this, we first perform a rigorous probabilistic analysis to derive lower bounds on detection and attribution performance for any given set of user watermarks. Then, we design an efficient algorithm to select user watermarks to maximize these lower bounds, thereby optimizing the detection and attribution performance.

## 5 Detection and Attribution Performance

We first formally define three key metrics to evaluate the performance of detection and attribution. We demonstrate in Appendix B that other relevant performance metrics can be derived from these three. Then, we theoretically analyze the evaluation metrics. All our proofs are shown in the Appendix.

**Content distributions:** Suppose we are given $s$ users $U = \{U_1, U_2, \cdots, U_s\}$, each of which has an unique watermark $w_i$, where $i = 1, 2, \cdots, s$. We denote the $s$ watermarks as a set $W = \{w_1, w_2, \cdots, w_s\}$. When a user $U_i$ generates content via the GenAI service, the service provider uses the encoder $E$ to embed the watermark $w_i$ into the content. We denote by $\mathcal{P}_i$ the probability distribution of the watermarked content generated by $U_i$. Note that two users $U_i$ and $U_j$ may have different AI-generated, watermarked content distributions $\mathcal{P}_i$ and $\mathcal{P}_j$. This is because the two users have different watermarks and they may be interested in generating different types of content. Moreover, we denote by $\mathcal{Q}$ the probability distribution of non-AI-generated content.

## 5.1 Evaluation Metrics

**(User-dependent) True Detection Rate (TDR):** TDR is the probability that an AI-generated content is correctly detected. Note that different users may have different AI-generated content distributions. Therefore, TDR depends on users. We denote by $TDR_i$ the true detection rate for the watermarked content generated by user $U_i$, i.e., $TDR_i$ is the probability that the content $C$ sampled from $\mathcal{P}_i$ uniformly at random is correctly detected as AI-generated. Formally, we have: $TDR_i = \Pr_{C \sim \mathcal{P}_i}(\max_{j \in \{1,2,\cdots,s\}} BA(D(C), w_j) \geq \tau)$, where the notation $\sim$ indicates the content is sampled from a distribution uniformly at random.

**False Detection Rate (FDR):** FDR is the probability that the content $C$ sampled from the non-AI-generated content distribution $\mathcal{Q}$ uniformly at random is detected as AI-generated. Note that FDR does not depend on users. Formally, we have: $FDR = \Pr_{C \sim \mathcal{Q}}(\max_{j \in \{1,2,\cdots,s\}} BA(D(C), w_j) \geq \tau)$.

**(User-dependent) True Attribution Rate (TAR):** TAR is the probability that an AI-generated content is correctly attributed to the user that generated the content. Like TDR, TAR also depends on users. We denote by $TAR_i$ the true attribution rate for the watermarked content generated by user $U_i$, i.e., $TAR_i$ is the probability that the content sampled from $\mathcal{P}_i$ uniformly at random is correctly attributed to user $U_i$. Formally, we have: $TAR_i = \Pr_{C \sim \mathcal{P}_i}(\max_{j \in \{1,2,\cdots,s\}} BA(D(C), w_j) \geq \tau \wedge BA(D(C), w_i) > \max_{j \in \{1,2,\cdots,s\}/\{i\}} BA(D(C), w_j))$.

The first term $\max_{j \in \{1,2,\cdots,s\}} BA(D(C), w_j) \geq \tau$ means that $C$ is detected as AI-generated, and the second term $BA(D(C), w_i) > \max_{j \in \{1,2,\cdots,s\}/\{i\}} BA(D(C), w_j)$ means that $C$ is attributed to user $U_i$. Note that attribution is only applied after detecting the content as AI-generated.

## 5.2 Formal Quantification of Watermarking

Intuitively, to theoretically analyze the detection and attribution performance (i.e., $TDR_i$, FDR, and $TAR_i$), we need a formal quantification of a watermarking method's behavior at decoding watermarks in AI-generated content and non-AI-generated content. Towards this end, we formally define $\beta$-*accurate* and $\gamma$-*random watermarking* in Appendix C.

**User-dependent $\beta_i$:** Since the users' AI-generated content may have different distributions $\mathcal{P}_i$, the same watermarking method may have different $\beta$ for different users, as illustrated in Figure 8 in the Appendix. To capture this phenomena, we consider the watermarking method is $\beta_i$-accurate for user $U_i$'s AI-generated content embedded with watermark $w_i$. Note that the same $\gamma$ is used across different users since it is used to characterize the behavior of the watermarking method for non-AI-generated content, which is user-independent. The parameters $\beta_i$ and $\gamma$ can be estimated using a set of AI-generated and non-AI-generated content, as shown in our experiments.

## 5.3 Detection and Attribution Performance

**Theorem 1** (Lower bound of $TDR_i$). *Suppose we are given $s$ users with any $s$ watermarks $W = \{w_1, w_2, \cdots, w_s\}$. When the watermarking method is $\beta_i$-accurate for user $U_i$'s AI-generated content, we have a lower bound of $TDR_i$:*

$$TDR_i \geq Pr(n_i \geq \tau n) + Pr(n_i \leq n - \tau n - \underline{\alpha_i} n), \tag{1}$$

*where $0.5 < \tau < \beta_i$, $\underline{\alpha_i} = \min_{j \in \{1,2,\cdots,s\}/\{i\}} BA(w_i, w_j)$, and $n_i \sim B(n, \beta_i)$ (binomial distribution).*

**Corollary 1.** *When the watermarking method is more accurate, the lower bound of $TDR_i$ is larger.*

**Theorem 2** (Upper bound of FDR). *Suppose we are given $s$ users with $s$ watermarks $W = \{w_1, w_2, \cdots, w_s\}$ and watermark $w_1$ is selected uniformly at random. We have an upper bound of FDR as follows:*

$$FDR \leq Pr(n_1 \geq \tau n) + Pr(n_1 \leq n - \tau n + \overline{\alpha_1} n), \tag{2}$$

*where $\overline{\alpha_1} = \max_{j \in \{2,3,\cdots,s\}} BA(w_1, w_j)$, and $n_1 \sim B(n, 0.5)$.*

Note that the upper bound of FDR in Theorem 2 does not depend on $\gamma$-random watermarking since we consider $w_1$ is picked uniformly at random. However, we found such upper bound is loose. This is because the second term of the upper bound considers the worst-case scenario of the $s$ watermarks. The next theorem shows that when the $s$ watermarks are constrained, in particular selected independently, we can derive a tighter upper bound of FDR.

**Theorem 3** (Alternative upper bound of FDR). *Suppose we are given $s$ users with $s$ watermarks $W = \{w_1, w_2, \cdots, w_s\}$ selected independently. When the watermarking method is $\gamma$-random for non-AI-generated content, we have an upper bound of FDR as follows:*

$$FDR \leq 1 - Pr(n' < \tau n)^s, \tag{3}$$

*where $n' \sim B(n, 0.5 + \gamma)$.*

**Corollary 2.** *When the watermarking method is more random for non-AI-generated content, i.e., $\gamma$ is closer to 0, the upper bound of FDR is smaller.*

**Theorem 4** (Lower bound of TAR$_i$). *Suppose we are given $s$ users with any $s$ watermarks $W = \{w_1, w_2, \cdots, w_s\}$. When the watermarking method is $\beta_i$-accurate for user $U_i$'s AI-generated content, we have a lower bound of TAR$_i$ as follows:*

$$TAR_i \geq Pr(n_i \geq \max\{\lfloor \frac{1 + \overline{\alpha_i}}{2} n \rfloor + 1, \tau n\}), \tag{4}$$

*where $\overline{\alpha_i} = \max_{j \in \{1,2,\cdots,s\}/\{i\}} BA(w_i, w_j)$, and $n_i \sim B(n, \beta_i)$.*

Our Theorem 4 shows that the lower bound of TAR$_i$ is larger when $\beta_i$ is closer to 1, i.e., attribution performance is better when the watermarking method is more accurate. Moreover, the lower bound is larger when $\overline{\alpha_i}$ is smaller because it is easier to distinguish between users. This is a theoretical motivation on why our watermark selection problem aims to select watermarks for the users such that they have small pairwise bitwise accuracy. Due to page limits, we systematically analyze the impact of $s$ on detection bounds, and the difference between user-agnostic and user-aware detection in Appendix D.

**Detection implies attribution:** When $\tau > \frac{1+\overline{\alpha_i}}{2}$, the lower bound of TAR$_i$ in Theorem 4 becomes TAR$_i \geq Pr(n_i \geq \tau n)$. The second term of the lower bound of TDR$_i$ in Theorem 1 is usually much smaller than the first term. In other words, the lower bound of TDR$_i$ is also roughly $Pr(n_i \geq \tau n)$. Therefore, when $\tau$ is large enough (i.e., $> \frac{1+\overline{\alpha_i}}{2}$), TDR$_i$ and TAR$_i$ are very close, which is also confirmed in our experiments. This result indicates that once the AI-generated content is correctly detected, it would also be correctly attributed.

# 6 SELECTING WATERMARKS FOR USERS

Our Theorems 1 and 4 reveal that detection and attribution performance improves when users' watermarks are more dissimilar. Leveraging this theoretical insight, we formulate the watermark selection problem, which aims to assign the most dissimilar watermarks to users. However, we demonstrate that this problem is NP-hard by reducing the well-known farthest string problem to it. This NP-hardness highlights the difficulty of developing an efficient exact solution. Consequently, we propose an efficient approximate solution to address the challenge.

## 6.1 FORMULATING A WATERMARK SELECTION PROBLEM

Intuitively, if two users have similar watermarks, then it is hard to distinguish between them for the attribution. An extreme example is that two users have the same watermark, making it impossible to attribute either of them. In fact, our theoretical analysis in Section 5 shows that attribution performance is better if the maximum pairwise bitwise accuracy between the users' watermarks is smaller. Thus, to enhance attribution, we aim to select watermarks for the $s$ users to minimize their maximum pairwise bitwise accuracy. Formally, we formulate watermark selection as the following optimization problem: $\min_{w_1, w_2, \cdots, w_s} \max_{i,j \in \{1,2,\cdots,s\}, i \neq j} BA(w_i, w_j)$, where $BA$ stands for bitwise accuracy between two watermarks. This optimization problem jointly optimizes the $s$ watermarks simultaneously. As a result, it is very challenging to solve the optimization problem because the GenAI service provider does not know the number of registered users (i.e., $s$) in advance. In practice, users register in the GenAI service at very different times. To address the challenge, we select a watermark for a user at the time of his/her registration in the GenAI service. For the first user $U_1$, we select a watermark uniformly at random. Suppose we have selected watermarks for $s - 1$ users. Then, the $s$th user registers and we aim to select a watermark $w_s$ whose maximum bitwise accuracy with the existing $s - 1$ watermarks is minimized. Formally, we formulate a *watermark selection problem* as follows:

$$\min_{w_s} \max_{i \in \{1,2,\cdots,s-1\}} BA(w_i, w_s). \tag{5}$$

### 6.2 SOLVING THE WATERMARK SELECTION PROBLEM

**NP-hardness:** We can show that our watermark selection problem in Equation 5 is NP-hard. In particular, we can reduce the well-known *farthest string problem* (Lanctot et al., 2003), which is NP-hard, to our watermark selection problem. In the farthest string problem, we aim to find a string that is the farthest from a given set of strings. We can view a string as a watermark in our watermark selection problem, the given set of strings as the watermarks of the $s-1$ users, and the similarity metric between two strings as our bitwise accuracy. Then, we can reduce the farthest string problem to our watermark selection problem, which means that our watermark selection problem is also NP-hard. This NP-hardness implies that it is very challenging to develop an efficient exact solution for our watermark selection problem. We note that efficiency is important for watermark selection as we aim to select a watermark for a user at the time of registration. Thus, we aim to develop an *efficient* algorithm that *approximately* solves the watermark selection problem.

**Decision problem:** To develop an efficient algorithm to approximately solve our watermark selection problem, we first define its *decision problem*. Specifically, given the maximum number of matched bits between $w_s$ and the existing $s-1$ watermarks as $m$, the decision problem aims to find such a $w_s$ if there exists one and return *NotExist* otherwise. Formally, the decision problem is to find any watermark $w_s$ in the following set if the set is nonempty: $w_s \in \{w | \max_{i \in \{1,2,\cdots,s-1\}} BA(w_i, w) \leq m/n\}$, where $n$ is the watermark length. Next, we discuss how to solve the decision problem and then turn the algorithm to solve our watermark selection problem.

**Approximate bounded search tree algorithm (A-BSTA):** We adapt BSTA (Gramm et al., 2003) as an efficient approximate solution to our decision problem. The details of BSTA and other baselines such as NRG (Chen et al., 2016) can be found in Appendix K. Specifically, A-BSTA makes two adaptions of BSTA. First, we constrain the recursion depth $d$ to be a constant (e.g., 8 in our experiments) instead of $m$, which makes the algorithm approximate but improves the efficiency substantially. Second, instead of initializing $w_s$ as $\neg w_1$, we initialize $w_s$ as an uniformly random watermark. As Table 3 in the Appendix shows, our initialization further improves the performance of A-BSTA. This is because a random initialization is more likely to have small bitwise accuracy with all existing watermarks. Note that BSTA, NRG, and A-BSTA all return *NotExist* if they cannot find a solution $w_s$ to the decision problem.

**Solving our watermark selection problem:** Given an algorithm (e.g., BSTA, NRG, or A-BSTA) to solve the decision problem, we turn it as a solution to our watermark selection problem. Specifically, our idea is to start from a small $m$, and then solve the decision problem. If we cannot find a watermark $w_s$ for the given $m$, we increase it by 1 and solve the decision problem again. We repeat this process until finding a watermark $w_s$. Note that we start from $m = \max_{i \in \{1,2,\cdots,s-2\}} n \cdot BA(w_i, w_{s-1})$, i.e., the maximum number of matched bits between $w_{s-1}$ and the other $s-2$ watermarks. This is because an $m$ smaller than this value is unlikely to produce a watermark $w_s$ as it failed to do so when selecting $w_{s-1}$. Algorithm 3 in the Appendix shows our method.

## 7 EXPERIMENTS

**Datasets:** We consider both AI-generated and non-AI-generated images. For AI-generated, we use three public datasets (Wang et al., 2023; Turc & Nemade, 2022; Images, 2023) generated respectively by Stable Diffusion, Midjourney, and DALL-E 2. For each dataset, we sample 10,000 images for training watermark encoders and decoders; and we sample 1,000 images for testing. For non-AI-generated, we combine the images in COCO (Lin et al., 2014), ImageNet (Deng et al., 2009), and Conceptual Caption (Sharma et al., 2018), and sample 1,000 images from the combined set uniformly at random as our non-AI-generated dataset. We scale the image size in all datasets to be $128 \times 128$.

**Watermarking methods:** We consider three representative image watermarking methods: HiDDeN (Zhu et al., 2018), StegaStamp (Tancik et al., 2020), and PRC watermark (Gunn et al., 2025). HiDDeN serves as the foundation for modern deep learning–based watermarking approaches. StegaStamp is a state-of-the-art method that embeds a watermark into an image after it has been generated. PRC watermark, on the other hand, represents a state-of-the-art approach where watermarks are inherently embedded during the image generation process. For HiDDeN and StegaStamp, we train the encoder–decoder pair for each GenAI model using its corresponding AI-generated image training set, and evaluate performance on the testing set.

**Evaluation metrics:** We use TDR, FDR, and TAR. FDR is the fraction of the 1,000 non-AI-generated images that are falsely detected as AI-generated. For each user $U_i$, we embed its watermark into 100 images randomly sampled from a testing AI-generated image dataset; and then we calculate the $\text{TDR}_i$ and $\text{TAR}_i$ for the user. In most of our experiments, we report the *average TDR* and *average TAR*, which respectively are the average $\text{TDR}_i$ and $\text{TAR}_i$ among the $s$ users. However, average TDR and average TAR cannot reflect the detection/attribution performance for the worst-case users, i.e., some users may have quite small $\text{TDR}_i$/$\text{TAR}_i$, but the average TDR/TAR may still be very large. Therefore, we further consider the 1% users (at least 1 user) with the smallest $\text{TDR}_i$ (or $\text{TAR}_i$) and report their average TDR (or TAR), which we call *worst 1% TDR* (or *worst 1% TAR*).

**Parameter settings:** By default, we set $s = 100,000$ (due to limited computation resource), $n = 64$, and $\tau = 0.9$. We also explore $s = 1,000,000$ in one of our experiments. Unless otherwise mentioned, we show results for the Stable Diffusion dataset.

## 7.1 DETECTION AND ATTRIBUTION RESULTS IN DIFFERENT SCENARIOS

We evaluate our method's detection and attribution performance across three scenarios: in the absence of post-processing, under common post-processing, and under adversarial post-processing.

**Without post-processing:** We first show results when the AI-generated, watermarked images are not post-processed. For each GenAI model, we compute the TDR/TAR of each user and the FDR. The FDRs for the three GenAI models are nearly 0. Then, we rank the users' TARs (or TDRs) in a non-descending order. Figure 2 presents the ranked TARs of 100,000 users for the three GenAI models under HiDDeN, StegaStamp, and PRC. We only evaluate PRC on Stable Diffusion, since this method relies on the inverse diffusion process, which is only feasible with the open-source model. Note that the curve of TDR overlaps with that of TAR and thus is omitted in the figure for simplicity. TDR and TAR overlap because $\tau = 0.9 > \frac{1+\overline{\alpha_i}}{2}$ (0.89 in our experiments), which is consistent with our theoretical analysis in Appendix D that shows detection implies attribution in such settings. Our results show that watermark-based detection and attribution are accurate when the AI-generated, watermarked images are not post-processed. Specifically, the worst $\text{TAR}_i$ or $\text{TDR}_i$ is larger than 0.94; less than 0.1% of users have $\text{TAR}_i$/$\text{TDR}_i$ smaller than 0.98; and 85% of users have $\text{TAR}_i$/$\text{TDR}_i$ of 1 for Midjourney and DALL-E 2, and 60% of such users for Stable Diffusion.

**Impact of $s$, $n$, and $\tau$:** Figure 6 in the Appendix shows the average TDR/TAR, worst 1% TDR/TAR, and FDR when $s$, $n$, or $\tau$ varies. Both average TDR and TAR are close to 1, and FDR is close to 0, as $s$ varies from 10 to 1,000,000. The average TDR/TAR slightly decrease when $n$ increases from 64 to 80, while the worst 1% TDR/TAR slightly increases as $n$ increases from 32 to 48 and then decreases

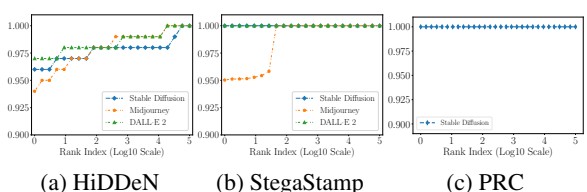

(a) HiDDeN     (b) StegaStamp     (c) PRC

Figure 2: Ranked TARs of the 100,000 users.

as $n$ further increases. Our result implies that HiDDeN may be unable to accurately encode/decode long watermarks. As $\tau$ increases, average TDR/TAR decrease, and FDR also decreases. Such trade-off of $\tau$ aligns with Theorem 1, 3, and 4. More details can be found in Appendix L.

**Common post-processing:** Common post-processing is often used to evaluate the robustness of watermarking in *non-adversarial settings*. We use JPEG, Gaussian noise, Gaussian blur, and Brightness/Contrast, whose details are shown in Appendix M. We use adversarial training to train HiDDeN and the training details can be found in Appendix M. Figure 9 in the Appendix show the detection/attribution results when a common post-processing method with different parameters is applied to the (AI-generated and non-AI-generated) images. Results also show the average SSIM (Wang et al., 2004) between a (AI-generated and non-AI-generated) image and its post-processed version. Our results show that detection and attribution are robust to common post-processing. In particular, the average TDR and TAR are still high when a common post-processing does not sacrifice image quality substantially. For instance, average TDR and TAR start to decrease sharply when the quality factor $Q$ of JPEG is smaller than 40. However, the average SSIM between watermarked images and their post-processed versions also drops quickly. Figure 10 in the Appendix shows a watermarked image and the versions post-processed by different methods.

**Adversarial post-processing:** Adversarial post-processing (Jiang et al., 2023b) carefully perturbs a watermarked image to evade detection/attribution. Watermarking is not robust to adversarial post-processing in white-box setting. Thus, detection/attribution is also not robust in such setting, i.e., TDR/TAR can be reduced to 0 while maintaining image quality. Figure 11 in the Appendix shows the average SSIM between watermarked images and their adversarially post-processed versions in the black-box setting (i.e., using attack WEvade-B-Q (Jiang et al., 2023b)) as a function of the number of queries to the detection API for *each* watermarked image, where the watermarking method is HiDDeN. Both TDR and TAR are 0 in these experiments since WEvade-B-Q always guarantees evasion. However, adversarial post-processing substantially sacrifices image quality in the black-box setting (i.e., SSIM is small) even if an attacker can query the detection API for a large number of times. Figure 12 in the Appendix shows several examples. Our results show that detection/attribution have good robustness to adversarial post-processing in the black-box setting.

### 7.2 Comparing Watermark Selection Methods

We compare three watermark selection methods: Random, NRG (Chen et al., 2016), and A-BSTA. NRG is the state-of-the-art approximate algorithm to the farthest string problem and we extend it to select watermarks (details in Appendix K). We do not use BSTA because it is not scalable, e.g., it takes more than 8 hours to select even 16 watermarks.

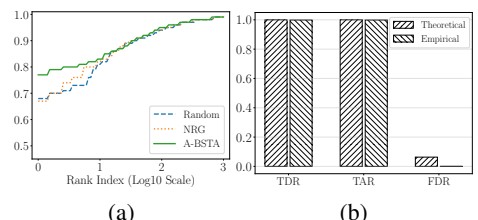

(a)  (b)

Figure 3: (a) Ranked $TAR_i$ of the worst 1K users for the three selection methods. (b) Theoretical vs. empirical results.

**Running time:** Table 4 in the Appendix shows the running time to generate a watermark averaged among the 100,000 watermarks. Although A-BSTA is slower than Random and NRG, the running time is acceptable.

**TAR:** Figure 3a shows the ranked TAR$_i$ of the worst 1,000 users, where the AI-generated images are post-processed by JPEG compression with quality factor $Q = 90$. The results indicate that A-BSTA outperforms NRG, which outperforms Random. This is because A-BSTA selects watermarks with smaller $\overline{\alpha_i}$, while Random selects watermarks with larger $\overline{\alpha_i}$ as shown in Figure 14 in the Appendix.

### 7.3 Theoretical vs. Empirical Results

We calculate the theoretical lower bounds of TDR$_i$ and TAR$_i$ of a user respectively using Theorem 1 and 4, while the theoretical upper bound of FDR using Theorem 3. We estimate $\beta_i$ as the bitwise accuracy between the decoded watermark and $w_i$ averaged among the testing AI-generated images, and estimate $\gamma$ using the fraction of bits in the decoded watermarks that are 1 among the non-AI-generated images. Figure 3b shows the average theoretical vs. empirical TDR/TAR, and theoretical vs. empirical FDR, when no post-processing is applied (Figure 15 in the Appendix shows the results when JPEG with $Q = 90$ is applied). The results show that our theoretical lower bounds of TDR and TAR match with empirical results well, which indicates that our derived lower bounds are tight. The theoretical upper bound of FDR is notably higher than the empirical FDR. This is because some bits may have larger probabilities to be 1 or 0 in the experiments, as shown in Figure 13 in the Appendix, but our theoretical analysis treats the bits equally, leading to a loose upper bound of FDR.

**Theoretical results when there are 100 millions users:** Due to limited computational resources, we show theoretical results on 100 million users in Table 5 in the Appendix, assuming $\beta_i = 0.99$, $\underline{\alpha_i} = 0.2$, $\gamma = 0.05$, and $\overline{\alpha_i} = 0.8$. We notice that TDR and TAR remain very close to 1.

## 8 Discussion

**Privacy issues:** Attribution systems inherently involve a trade-off between provenance and user privacy, since provenance requires tracing content back to a specific user, which naturally raises privacy concerns. Importantly, our watermark-based attribution does not require exposing user identity

or sensitive information to arbitrary third parties. The attribution metadata can be designed so that only trusted entities (e.g., the GenAI service provider or a designated verification authority, similar in role to the Public Key Infrastructure) are able to perform verification. This follows established practices in responsible provenance systems.

**Watermark forgery attack:** A malicious user may attempt to forge the watermark of a specific user for the given content based on existing watermarked content produced by that user, such that the forged content would be falsely attributed even though it was not actually generated by that user. According to state-of-the-art research on watermark forgery attacks, forgery in white-box settings, where the attacker has full access to the watermarking system, is easy but not very practical. However, in non-white-box settings, where the attacker has only limited knowledge of the watermarking system, forgery remains challenging even at the detection level, let alone for user-level attribution, since the malicious user does not know the specific user's watermark. For example, Steganalysis (Yang et al., 2024a) reports effectiveness against several content-agnostic watermarking methods, such as TreeRing (Wen et al., 2023), GaussianShading (Yang et al., 2024b), and RingID (Ci et al., 2024). However, it fails against many content-dependent state-of-the-art image watermarking methods such as Stable Signature (Fernandez et al., 2023), WAM (Sander et al., 2025), and PRC (Gunn et al., 2025), as shown in Table 6 in the Appendix.

**Using hash for watermark selection:** We also try using a hash function to generate user-specific watermarks (given the user ID). However, such a simple mechanism essentially picks watermarks for users randomly due to the pseudo randomness of hash functions. Therefore, this mechanism behaves similarly to our Random baseline (the details for Random and other baselines are in Appendix K), which generates a watermark uniformly at random for each user. Thus, both hash and Random baseline should only achieve sub-optimal attribution performance. Results are shown in Table 7 in the Appendix.

**Difference with steganography:** Steganography and watermarking both embed information into content, but they are designed for different goals. Steganography prioritizes secrecy and undetectability, whereas watermarking prioritizes robustness under post-processing. As a result, most steganographic schemes are not designed to withstand standard transforms (e.g., resizing, compression, filtering), which makes them unsuitable for attribution in real-world GenAI pipelines. In contrast, watermarking methods are explicitly designed for robustness and can survive such transformations, as demonstrated by our experiments.

**Influence of the number of content:** If multiple pieces of content are known to originate from the same user, they can indeed improve attribution accuracy. For example, one can perform attribution on each content independently and then aggregate the results (e.g., via majority vote). For example, using the HiDDeN watermark on the Stable Diffusion dataset with parameters $s = 100,000$, $n = 64$, and $\tau = 0.9$, our average TAR increases from 0.998 to 1.000 when majority vote aggregation is applied.

## 9 Conclusion and Future Work

We show that watermarks can be effectively used for user-aware detection and attribution of AI-generated content. The main challenge is how to select watermarks for users to maximize detection and attribution performance. We show that this can be addressed through a two-step solution. First, we theoretically derive the detection and attribution performance for any given set of user watermarks. Second, we select user watermarks to maximize the theoretical performance. Our empirical evaluation shows that, using the current state-of-the-art watermarking method, detection and attribution are highly accurate when the watermarked AI-generated content is either post-processed by common post-processing, or exposed to black-box adversarial post-processing with limited query budgets.

**Text watermarking:** Our theory and algorithm can be applied to text watermarking methods (Abdelnabi & Fritz, 2021) that use bitstrings as watermarks for attributing AI-generated text (see Appendix N for details). However, they may not be applicable to methods that do not rely on bitstring-based watermarks (Kirchenbauer et al., 2023). A promising direction for future work is to extend our framework to other modalities (Liu et al., 2024; Jiang et al., 2025), such as audio and video watermarking.

## 10 REPRODUCIBILITY STATEMENT

In this work, we make the following contributions: (1) we formally define the detection and attribution problem for AI-generated content; (2) we introduce metrics for evaluating detection and attribution performance; (3) we propose the notions of $\beta$-*accurate* and $\gamma$-*random* watermarking; and (4) we derive theoretical bounds for these evaluation metrics. We rigorously prove the corresponding theorems and validate the bounds, with complete proofs provided in the Appendix. For evaluation, we detail our parameter settings in Section 7, and our results can be reproduced using publicly available GitHub repositories. We will also release our code and datasets.

## ACKNOWLEDGMENTS

We thank the anonymous reviewers for their constructive comments. This work was supported by NSF grant No. 2450935, 2414406, 2125977, 2112562, 1937787.

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

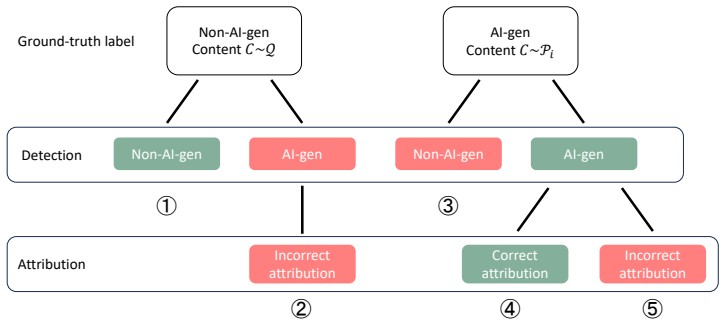

Figure 4: Taxonomy of detection and attribution results. Nodes with red color indicate incorrect detection/attribution.

## A USE OF LLMS

We use large language models to aid or polish writing at the sentence level, such as fixing grammar and re-wording sentences. LLMs were not involved in designing methods, conducting experiments, or drawing conclusions. No sensitive or proprietary data were shared with LLMs.

## B OTHER EVALUATION METRICS CAN BE DERIVED FROM $\text{TDR}_i$, FDR, AND $\text{TAR}_i$

We note that there are also other relevant detection and attribution metrics, e.g., the probability that the AI-generated content is incorrectly attributed to a user. We show that other relevant detection and attribution metrics can be derived from $\text{TDR}_i$, FDR, and $\text{TAR}_i$, and thus we focus on these three metrics in our work. Specifically, Figure 4 shows the taxonomy of detection and attribution results for non-AI-generated content and AI-generated content generated by user $U_i$. In the taxonomy trees, the first-level nodes represent ground-truth labels of content; the second-level nodes represent possible detection results; and the third-level nodes represent possible attribution results (attribution is performed only after the content is detected as AI-generated).

In the taxonomy trees, there are 5 branches in total, which are labeled as ①, ②, ③, ④, and ⑤ in the figure. Each branch starts from a root node and ends at a leaf node, and corresponds to a metric that may be of interest. For instance, our $\text{TDR}_i$ is the probability that the content $C \sim \mathcal{P}_i$ goes through branches ④ or ⑤; FDR is the probability that the content $C \sim \mathcal{Q}$ goes through branch ②; and $\text{TAR}_i$ is the probability that the content $C \sim \mathcal{P}_i$ goes through branch ④. The probability that the content goes through other branches can be calculated using $\text{TDR}_i$, FDR, and/or $\text{TAR}_i$. For instance, the probability that the non-AI-generated content $C \sim \mathcal{Q}$ is correctly detected as non-AI-generated is the probability that $C$ goes through the branch ①, which can be calculated as $1-$FDR. The probability that the AI-generated content $C \sim \mathcal{P}_i$ is incorrectly detected as non-AI-generated is the probability that $C$ goes through the branch ③, which can be calculated as $1-\text{TDR}_i$. The probability that a user $U_i$'s AI-generated content $C \sim \mathcal{P}_i$ is correctly detected as AI-generated but incorrectly attributed to a different user $U_j$ is the probability that $C$ goes through the branch ⑤, which can be calculated as $\text{TDR}_i-\text{TAR}_i$.

## C DEFINITIONS OF $\beta$-ACCURATE AND $\gamma$-RANDOM WATERMARKING

**Definition 3** ($\beta$-accurate watermarking). For the randomly sampled AI-generated content $C \sim \mathcal{P}$ embedded with watermark $w$, the bits of the decoded watermark $D(C)$ are independent and each bit matches with that of $w$ with probability $\beta$. Formally, we have $Pr(D(C)[k] = w[k]) = \beta$, where $C \sim \mathcal{P}$, $D$ is the decoder, and $[k]$ represents the $k$th bit of a watermark. We say a watermarking method is $\beta$-accurate if it satisfies the above condition.

**Definition 4** ($\gamma$-random watermarking). For the randomly sampled non-AI-generated content $C \sim \mathcal{Q}$ without any watermark embedded, the bits of the decoded watermark $D(C)$ are independent and

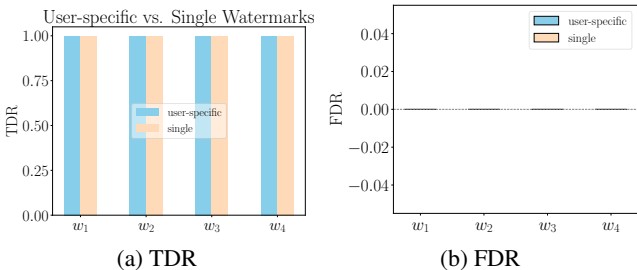

Figure 5: User-agnostic vs. user-aware results. $w_1$, $w_2$, $w_3$, and $w_4$ are 4 different random watermarks for the user-agnostic setting, where $s$=100,000 users for the user-aware setting. Results show that the two settings achieve comparable TDR and FDR.

each bit is 1 with probability at least $0.5 - \gamma$ and at most $0.5 + \gamma$, where $\gamma \in [0, 0.5]$. Formally, we have $|Pr(D(C)[k] = 1) - 0.5| \leq \gamma$, where $C \sim \mathcal{Q}$ and $[k]$ represents the $k$th bit of a watermark. We say a watermarking method is $\gamma$-random if it satisfies the above condition.

The parameter $\beta$ is used to characterize the accuracy of the watermarking method at encoding/decoding a watermark in AI-generated content. In particular, the watermarking method is more accurate when $\beta$ is closer to 1. For a $\beta$-accurate watermarking method, the number of matched bits between the decoded watermark $D(C)$ for the watermarked content $C$ and the ground-truth watermark follows a binomial distribution with parameters $n$ and $\beta$, where $n$ is the watermark length.

The parameter $\gamma$ characterizes the behavior of the watermarking method for non-AI-generated content. In particular, the decoded watermark for the non-AI-generated (i.e., unwatermarked) content is close to a uniformly random watermark, where $\gamma$ quantifies the difference between them. The watermarking method is more random for non-AI-generated content if $\gamma$ is closer to 0.

**Incorporating post-processing:** Our definition of $\beta$-accurate and $\gamma$-random watermarking can also incorporate post-processing (e.g., JPEG compression) that an attacker/user may apply to AI-generated or non-AI-generated content. In particular, we can replace $D(C)$ as $D(P(C))$ in our definitions, where $P$ stands for post-processing of the content $C$. When AI-generated content is post-processed, the watermarking method may become less accurate, i.e., $\beta$ may decrease.

## D  THEORETICAL ANALYSIS

**Impact of $s$ on the bounds:** Intuitively, when there are more users, i.e., $s$ is larger, it is more likely to have at least one user whose watermark has a bitwise accuracy with the decoded watermark $D(C)$ that is no smaller than $\tau$. As a result, both $\text{TDR}_i$ and FDR may increase as $s$ increases, i.e., $s$ controls a trade-off between $\text{TDR}_i$ and FDR. Our theoretical results align with this intuition. On one hand, our Theorem 1 shows that the lower bound of $\text{TDR}_i$ is larger when $s$ is larger. In particular, when $s$ increases, the parameter $\underline{\alpha_i}$ may become smaller. Therefore, the second term of the lower bound increases, leading to a larger lower bound of $\text{TDR}_i$. On the other hand, the upper bound of FDR in both Theorem 2 and Theorem 3 increases as $s$ increases. In particular, in Theorem 2, the parameter $\overline{\alpha_1}$ becomes larger when $s$ increases, leading to a larger second term of the upper bound.

**User-agnostic vs. user-aware detection:** Existing watermark-based detection is user-agnostic, i.e., it does not distinguish between different users when embedding a watermark into the AI-generated content. The first term of the lower bound in our Theorem 1 is a lower bound of TDR for user-agnostic detection; the first term of the upper bound in our Theorem 2 is an upper bound of FDR for user-agnostic detection; and the upper bound with $s = 1$ in our Theorem 3 is an alternative upper bound of FDR for user-agnostic detection. Therefore, compared to user-agnostic detection, our user-aware detection achieves larger TDR but also larger FDR. Figure 5 illustrates empirical TDR and FDR results for user-agnostic and user-aware detection.

# E   Proof of Theorem 1

Intuitively, a user $U_i$'s AI-generated content $C \sim \mathcal{P}_i$ can be correctly detected as AI-generated in two cases:

- **Case I.** The decoded watermark $D(C)$ is similar enough to the user $U_i$'s watermark $w_i$.
- **Case II.** The decoded watermark $D(C)$ is dissimilar to $w_i$ but similar enough to some other user's watermark.

Case II is more likely to happen when $w_i$ is more dissimilar to some other user's watermark, i.e., when $\underline{\alpha_i} = \min_{j \in \{1,2,\cdots,s\}/\{i\}} BA(w_i, w_j)$ is smaller. This is because the fact that $D(C)$ is dissimilar to $w_i$ and $w_i$ is dissimilar to some other user's watermark implies that $D(C)$ is similar to some other user's watermark. Formally, we can derive a lower bound of $\text{TDR}_i$ as follows:

For $C \sim \mathcal{P}_i$, we denote $w = D(C)$, $n_i = BA(w, w_i)n$, and $n_j = BA(w, w_j)n$ for $j \in \{1, 2, \cdots, s\}/\{i\}$. Then we have the following:

$$|w - \neg w_i|_1 = n_i,$$
$$|\neg w_i - w_j|_1 = BA(w_i, w_j)n,$$
$$|w - w_j|_1 = n - n_j,$$

where $\neg w_i$ means flipping each bit of the watermark $w_i$, $|\cdot|_1$ is $\ell_1$ distance between two binary vectors. According to the triangle inequality, we have:

$$|w - w_j|_1 \leq |w - \neg w_i|_1 + |\neg w_i - w_j|_1$$
$$= n_i + BA(w_i, w_j)n.$$

Therefore, we derive the lower bound of $n_j$ for $j \in \{1, 2, \cdots, s\}/\{i\}$ as follows:

$$n_j = n - |w - w_j|_1$$
$$\geq n - n_i - BA(w_i, w_j)n.$$

Thus, we derive the lower bound of $\text{TDR}_i$ as follows:

$$
\begin{aligned}
TDR_i &= 1 - \Pr(n_i < \tau n \wedge \max_{j \in \{1,2,\cdots,s\}/\{i\}} n_j < \tau n)) \\
&\geq 1 - \Pr(n_i < \tau n \wedge \max_{j \in \{1,2,\cdots,s\}/\{i\}} n - n_i - BA(w_i, w_j)n < \tau n)) \\
&= 1 - \Pr(n_i < \tau n \wedge n - n_i - \underline{\alpha_i} n < \tau n) \\
&= 1 - \Pr(n - \tau n - \underline{\alpha_i} n < n_i < \tau n) \\
&= \Pr(n_i \geq \tau n) + \Pr(n_i \leq n - \tau n - \underline{\alpha_i} n),
\end{aligned}
$$

where $n_i \sim B(n, \beta_i)$ and $\underline{\alpha_i} = \min_{j \in \{1,2,\cdots,s\}/\{i\}} BA(w_i, w_j)$.

The two terms in the lower bound respectively bound the probabilities for Case I and Case II of correctly detecting user $U_i$'s AI-generated content.

# F   Proof of Corollary 1

According to Theorem 1, the lower bound of $\text{TDR}_i$ is $1 - \Pr(n - \tau n - \underline{\alpha_i} n < n_i < \tau n)$. For an integer $r \in (n - \tau n - \underline{\alpha_i} n, \tau n)$ and $n_i \sim B(n, \beta_i)$, we have the following:

$$\Pr(n_i = r) = \binom{n}{r} \beta_i^r (1 - \beta_i)^{n-r}.$$

Then we compute the partial derivative of the probability with respect to $\beta_i$ as follows:

$$
\frac{\partial \Pr(n_i = r)}{\partial \beta_i} = \binom{n}{r} \beta_i^{r-1} (1 - \beta_i)^{n-r-1} (r(1 - \beta_i) - (n - r)\beta_i)
$$
$$
< \binom{n}{r} \beta_i^{r-1} (1 - \beta_i)^{n-r-1} (\tau - \beta_i)n.
$$

The partial derivative is smaller than 0 when $\tau < \beta_i$. Therefore, the probability $\Pr(n_i = r)$ decreases as $\beta_i$ increases for any integer $r \in (n - \tau n - \underline{\alpha_i} n, \tau n)$. Thus, the lower bound of $\text{TDR}_i$ increases as $\beta_i$ becomes closer to 1.

## G    PROOF OF THEOREM 2

Intuitively, the non-AI-generated content $C \sim \mathcal{Q}$ is also incorrectly detected as AI-generated in two cases: 1) the decoded watermark $D(C)$ is similar enough with some user's watermark, e.g., $w_1$; and 2) $D(C)$ is dissimilar to $w_1$ but similar enough to some other user's watermark. Based on this intuition, we can derive an upper bound of FDR as follows:

For $C \sim Q$, we denote $n_1 = BA(D(C), w_1)n$ and $n_j = BA(D(C), w_j)n$ for $j \in \{1, 2, \cdots, s\}$. Then, we have the following:

$$
\begin{aligned}
FDR &= 1 - \Pr(\max_{j \in \{1,2,\cdots,s\}} n_j < \tau n) \\
&= 1 - \Pr(n_1 < \tau n \wedge \max_{j \in \{2,3,\cdots,s\}} n_j < \tau n).
\end{aligned}
$$

To derive an upper bound of FDR, we denote:

$$
\begin{aligned}
|w - w_1|_1 &= n - n_1, \\
|w_1 - w_j|_1 &= n - BA(w_1, w_j)n, \\
|w - w_j|_1 &= n - n_j.
\end{aligned}
$$

According to the triangle inequality, we have the following:

$$
\begin{aligned}
|w - w_j|_1 &\geq |w_1 - w_j|_1 - |w - w_1|_1 \\
&= n_1 - BA(w_1, w_j)n.
\end{aligned}
$$

Therefore, we derive the upper bound of $n_j$ for $j \in \{2, 3, \cdots, s\}$ as follows:

$$
\begin{aligned}
n_j &= n - |w - w_j|_1 \\
&\leq n - n_1 + BA(w_1, w_j)n.
\end{aligned}
$$

Thus, we derive the upper bound of FDR as follows:

$$
\begin{aligned}
FDR &= 1 - \Pr(n_1 < \tau n \wedge \max_{j \in \{2,3,\cdots,s\}} n_j < \tau n) \\
&\leq 1 - \Pr(n_1 < \tau n \wedge \max_{j \in \{2,3,\cdots,s\}} n - n_1 + BA(w_1, w_j)n < \tau n)) \\
&= 1 - \Pr(n_1 < \tau n \wedge n - n_1 + \overline{\alpha_1} n < \tau n)) \\
&= 1 - \Pr(n - \tau n + \overline{\alpha_1} n < n_1 < \tau n) \\
&= \Pr(n_1 \geq \tau n) + \Pr(n_1 \leq n - \tau n + \overline{\alpha_1} n),
\end{aligned}
$$

where $n_1 \sim B(n, 0.5)$ and $\overline{\alpha_1} = \max_{j \in \{2,3,\cdots,s\}} BA(w_1, w_j)$.

## H    PROOF OF THEOREM 3

For $C \sim Q$, we denote $n_j = BA(D(C), w_j)n$ for $j \in \{1, 2, \cdots, s\}$, and we have the following:

$$
\begin{aligned}
FDR &= 1 - \Pr(\max_{j \in \{1,2,\cdots,s\}} n_j < \tau n) \\
&= 1 - \prod_{j \in \{1,2,\cdots,s\}} \Pr(n_j < \tau n).
\end{aligned}
$$

According to Definition 4, for any $k \in \{1, 2, \cdots, n\}$ and any $j \in \{1, 2, \cdots, s\}$, the decoding of each bit is independent and the probability that $D(C)[k]$ matches with $w_j[k]$ is at most $0.5 + \gamma$ no matter $w_j[k]$ is 1 or 0. Therefore, we have the following:

$$
\begin{aligned}
FDR &= 1 - \prod_{j \in \{1,2,\cdots,s\}} \Pr(n_j < \tau n) \\
&\leq 1 - \Pr(n' < \tau n)^s,
\end{aligned}
$$

where $n'$ follows the binomial distribution with parameters $n$ and $0.5 + \gamma$, i.e., $n' \sim B(n, 0.5 + \gamma)$.

## I    PROOF OF COROLLARY 2

According to Theorem 3, the probability $Pr(n' < \tau n)$ increases when $\gamma$ decreases. Therefore, the upper bound of FDR decreases as $\gamma$ becomes closer to 0.

## J    PROOF OF THEOREM 4

Suppose we are given a user $U_i$'s AI-generated content $C \sim \mathcal{P}_i$. Intuitively, if the watermark $w_i$ is very dissimilar to the other $s-1$ watermarks, i.e., $\overline{\alpha_i} = \max_{j \in \{1,2,\cdots,s\}/\{i\}} BA(w_i, w_j)$ is small, then $C$ can be correctly attributed to $U_i$ once $C$ is detected as AI-generated, i.e., the decoded watermark $D(C)$ is similar enough to $w_i$. If the watermark $w_i$ is similar to some other watermark, i.e., $\overline{\alpha_i}$ is large, then the decoded watermark $D(C)$ has to be very similar to $w_i$ in order to correctly attribute $C$ to $U_i$. Formally, we can derive a lower bound of $\text{TAR}_i$.

For $C \sim \mathcal{P}_i$, we denote $w = D(C)$, $n_i = BA(w, w_i)n$, and $n_j = BA(w, w_j)n$ for $j \in \{1, 2, \cdots, s\}$. Then we have the following:

$$|w - \neg w_i|_1 = n_i,$$
$$|\neg w_i - w_j|_1 = BA(w_i, w_j)n,$$
$$|w - w_j|_1 = n - n_j.$$

According to the triangle inequality, we have:

$$|w - w_j|_1 \geq |w - \neg w_i|_1 - |\neg w_i - w_j|_1$$
$$= n_i - BA(w_i, w_j)n.$$

Therefore, we derive the upper bound of $n_j$ for $j \in \{1, 2, \cdots, s\}/\{i\}$ as follows:

$$n_j = n - |w - w_j|_1$$
$$\leq n - n_i + BA(w_i, w_j)n.$$

Thus, we derive the lower bound of $\text{TAR}_i$ as follows:

$$
\begin{aligned}
TAR_i =& \Pr(\max_{j \in \{1,2,\cdots,s\}} n_j \geq \tau n \wedge n_i > \max_{j \in \{1,2,\cdots,s\}/\{i\}} n_j) \\
\geq& \Pr(\max_{j \in \{1,2,\cdots,s\}} n_j \geq \tau n \wedge n_i > \max_{j \in \{1,2,\cdots,s\}/\{i\}} n - n_i + BA(w_i, w_j)n) \\
=& \Pr(\max_{j \in \{1,2,\cdots,s\}} n_j \geq \tau n \wedge n_i > \frac{n + \overline{\alpha_i}n}{2}) \\
=& \Pr(\max_{j \in \{1,2,\cdots,s\}} n_j \geq \tau n \wedge n_i > \frac{n + \overline{\alpha_i}n}{2} \mid n_i \geq \tau n) \cdot \Pr(n_i \geq \tau n) \\
&+ \Pr(\max_{j \in \{1,2,\cdots,s\}} n_j \geq \tau n \wedge n_i > \frac{n + \overline{\alpha_i}n}{2} \mid n_i < \tau n) \cdot \Pr(n_i < \tau n) \\
\geq& \Pr(n_i > \frac{n + \overline{\alpha_i}n}{2} \mid n_i \geq \tau n) \cdot \Pr(n_i \geq \tau n) \\
=& \Pr(n_i > \frac{n + \overline{\alpha_i}n}{2} \wedge n_i \geq \tau n) \\
=& \Pr(n_i \geq \max\{\lfloor \frac{1 + \overline{\alpha_i}}{2}n \rfloor + 1, \tau n\}),
\end{aligned}
$$

where $n_i \sim B(n, \beta_i)$ and $\overline{\alpha_i} = \max_{j \in \{1,2,\cdots,s\}/\{i\}} BA(w_i, w_j)$.

## K    WATERMARK SELECTION ALGORITHMS

**Random:**  The most straightforward method to approximately solve the watermark selection problem in Equation 5 is to generate a watermark uniformly at random as $w_s$. We denote this method as *Random*. The limitation of this method is that the selected watermark $w_s$ may be very similar to some existing watermarks, i.e., $\max_{i \in \{1,2,\cdots,s-1\}} BA(w_i, w_s)$ is large, making attribution less accurate, as shown in our experiments.

---

**Algorithm 1** BSTA$(w_s, d, m)$

---

**Input:** Initial watermark $w_s$, recursion depth $d$, and $m$.
**Output:** $w_s$ or NotExist.
1: **if** $d < 0$ **then**
2:     return $NotExist$
3: **end if**
4: $i^* \leftarrow \arg\max_{i \in \{1,2,\cdots,s-1\}} BA(w_i, w_s)$
5: **if** $BA(w_{i^*}, w_s) > (m+d)/n$ **then**
6:     return $NotExist$
7: **else if** $BA(w_{i^*}, w_s) \leq m/n$ **then**
8:     return $w_s$
9: **end if**
10: $B \leftarrow \{k | w_s[k] = w_{i^*}[k], k = 1, 2, \cdots, n\}$
11: Choose any $B' \subset B$ with $|B'| = m + 1$
12: **for all** $k \in B'$ **do**
13:     $w'_s \leftarrow w_s$
14:     $w'_s[k] \leftarrow \neg w'_s[k]$
15:     $w'_s \leftarrow$ BSTA$(w'_s, d-1, m)$
16:     **if** $w'_s$ is not $NotExist$ **then**
17:         return $w'_s$
18:     **end if**
19: **end for**
20: return $NotExist$

---

**Bounded search tree algorithm (BSTA) (Gramm et al., 2003):** Recall that our watermark selection problem is equivalent to the farthest string problem. Thus, our decision problem is equivalent to that of the farthest string problem, which has been studied extensively in the theoretical computer science community. In particular, BSTA is the state-of-the-art *exact* algorithm to solve the decision problem version of the farthest string problem. We apply BSTA to solve the decision problem version of our watermark selection problem exactly, which is shown in Algorithm 1. The key idea of BSTA is to initialize $w_s$ as $\neg w_1$ (i.e., each bit of $w_1$ flips), and then reduce the decision problem to a simpler problem recursively until it is easily solvable or there does not exist a solution $w_s$. In particular, given an initial $w_s$, BSTA first finds the existing watermark $w_{i^*}$ that has the largest bitwise accuracy with $w_s$. If $BA(w_{i^*}, w_s) \leq m/n$, then $w_s$ is already a solution to the decision problem and thus BSTA returns $w_s$. Otherwise, BSTA chooses any $m + 1$ bits that $w_s$ and $w_{i^*}$ match. For each of the chosen $m + 1$ bits, BSTA flips the corresponding bit in $w_s$ and recursively solves the decision problem using the new $w_s$ as an initialization. The recursion is applied $m$ times at most, i.e., the recursion depth $d$ is set as $m$ when calling Algorithm 1.

A key limitation of BSTA is that it has an exponential time complexity (Gramm et al., 2003). In fact, since the decision problem is NP-hard, all known *exact* solutions have exponential time complexity. Therefore, to enhance computation efficiency, we resort to approximate solutions. Next, we discuss the state-of-the-art approximate solution that adapts BSTA and a new approximate solution that we propose.

**Non Redundant Guess (NRG) (Chen et al., 2016):** Like BSTA, this approximate solution also first initializes $w_s$ as $\neg w_1$ and finds the existing watermark $w_{i^*}$ that has the largest bitwise accuracy with $w_s$. If $BA(w_{i^*}, w_s) \leq m/n$, then NRG returns $w_s$. Otherwise, NRG samples $n \cdot BA(w_{i^*}, w_s) - m$ bits that $w_s$ and $w_{i^*}$ match uniformly at random. Then, NRG flips these bits in $w_s$ and recursively solve the decision problem using the new $w_s$ as an initialization. Note that NRG stops the recursion when $m$ bits of the initial $w_s$ have been flipped. Algorithm 2 describes NRG.

**Approximate bounded search tree algorithm (A-BSTA):** The algorithm of our A-BSTA is shown as Algorithm 3. Note that binary search is another way to find a proper $m$. Specifically, we start with a small $m$ (denoted as $m_l$) that does not produce a $w_s$ and a large $m$ (denoted as $m_u$) that does produce a $w_s$. If $m = (m_l + m_u)/2$ produces a $w_s$, we update $m_u = (m_l + m_u)/2$; otherwise we update $m_l = (m_l + m_u)/2$. The search process stops when $m_l \geq m_u$. However, we found that increasing $m$ by 1 as in our Algorithm 3 is more efficient than binary search. This is because

---

**Algorithm 2** $NRG(w_s, m)$

---

**Input:** Initial watermark $w_s$ and $m$.
**Output:** $w_s$ or NotExist.
1: $F \leftarrow \emptyset$
2: $d \leftarrow m$
3: **while** $d > 0$ **do**
4:      $i^* \leftarrow \arg\max_{i \in \{1,2,\cdots,s-1\}} BA(w_i, w_s)$
5:      **if** $BA(w_{i^*}, w_s) > 2m/n$ **then**
6:          return $NotExist$
7:      **else if** $BA(w_{i^*}, w_s) \le m/n$ **then**
8:          return $w_s$
9:      **end if**
10:     $B \leftarrow \{k | w_s[k] = w_{i^*}[k] \wedge k \notin F, k = 1, 2, \cdots, n\}$
11:     $l \leftarrow n \cdot BA(w_{i^*}, w_s) - m$
12:     Sample $B' \subset B$ with $|B'| = l$ uniformly at random
13:     **for all** $k \in B'$ **do**
14:         $w_s[k] \leftarrow \neg w_s[k]$
15:     **end for**
16:     $d \leftarrow d - l$
17:     $F \leftarrow F \cup B'$
18: **end while**
19: return $NotExist$

---

**Algorithm 3** Solving our watermark selection problem

---

**Input:** Existing $s - 1$ watermarks $w_1, w_2, \cdots, w_{s-1}$.
**Output:** Watermark $w_s$.
1: $m \leftarrow \max_{i \in \{1,2,\cdots,s-2\}} n \cdot BA(w_i, w_{s-1})$
2: **while** $w_s$ is $NotExist$ **do**
3:     **if** BSTA **then**
4:         $w_s \leftarrow \neg w_1$
5:         $w_s \leftarrow BSTA(w_s, m, m)$
6:     **end if**
7:     **if** NRG **then**
8:         $w_s \leftarrow \neg w_1$
9:         $w_s \leftarrow NRG(w_s, m)$
10:    **end if**
11:    **if** A-BSTA **then**
12:        $w_s \leftarrow$ sampled uniformly at random
13:        $w_s \leftarrow BSTA(w_s, d, m)$
14:    **end if**
15:    **if** $w_s$ is $NotExist$ **then**
16:        $m \leftarrow m + 1$
17:    **end if**
18: **end while**
19: return $w_s$

---

increasing $m$ by 1 expands the search space of $w_s$ substantially, which often leads to a valid $w_s$. On the contrary, binary search would require solving the decision problem multiple times with different $m$ until finding that $m + 1$ is enough.

**Time complexity:** We analyze the time complexity of the algorithms to solve the decision problem. For Random, the time complexity is $O(n)$. For BSTA, the time complexity to solve the decision problem with parameter $m$ is $O(snm^m)$ according to (Gramm et al., 2003). For NRG, the time complexity is $O(sn + s\sqrt{m} \cdot 5^m)$ according to (Chen et al., 2016). For A-BSTA, the time complexity is $O(snm^d)$, where $d$ is a constant.

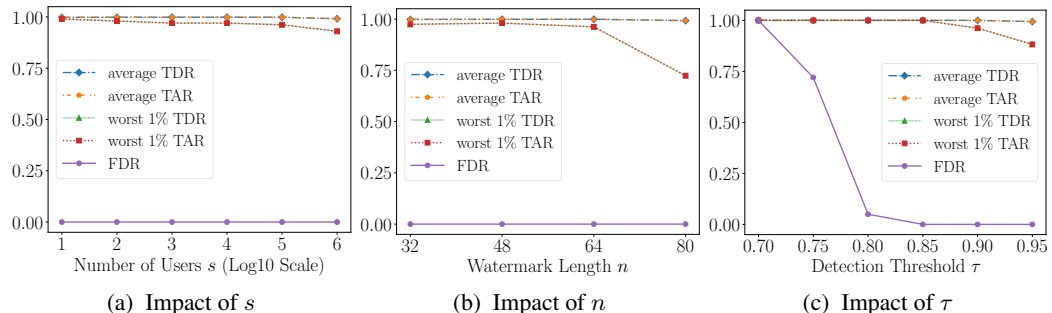

|   | (a) Impact of $s$ | (b) Impact of $n$ | (c) Impact of $\tau$ |
|---|---|---|---|

Figure 6: Impact of number of users $s$, watermark length $n$, and detection threshold $\tau$ on detection and attribution performance. The watermarking method is HiDDeN.

Table 1: The average $\beta_i$ of all users and of the worst 1% of users, where $\beta_i$ of a user is estimated using the testing images.

| Watermark Length $n$ | 32 | 48 | 64 | 80 |
|---|---|---|---|---|
| Average $\beta_i$ | 0.99 | 0.99 | 0.99 | 0.97 |
| Worst 1% $\beta_i$ | 0.92 | 0.97 | 0.90 | 0.84 |

## L  IMPACT OF $s$, $n$, AND $\tau$

**Impact of number of users $s$:** Figure 6a shows the average TDR, average TAR, worst 1% TDR, worst 1% TAR, and FDR when $s$ varies from 10 to 1,000,000. We have two observations. First, both average TDR and average TAR are consistently close to 1, and FDR is consistently close to 0, which means our detection and attribution are accurate. Second, worst 1% TDR and worst 1% TAR decrease as $s$ increases. This is because when there are more users, the worst 1% of them have smaller TDRs and TARs. Moreover, these worst 1% of users also make the average TDR and average TAR decrease slightly when $s$ increases from 100,000 to 1,000,000.

**Impact of watermark length $n$:** Figure 6b shows the average TDR, average TAR, worst 1% TDR, worst 1% TAR, and FDR when the watermark length $n$ varies from 32 to 80. The average TDR and average TAR slightly decrease when $n$ increases from 64 to 80, while the worst 1% TDR/TAR slightly increases as $n$ increases from 32 to 48 and then decreases as $n$ further increases. Table 1 shows the estimated average $\beta_i$ of all users and average $\beta_i$ of the worst 1% of users in $\beta$-accurate watermarking. We observe that the patterns of average TDR/TAR and worst 1% TDR/TAR are consistent with those of average $\beta_i$ and worst 1% $\beta_i$, respectively. These observations are consistent with our theoretical analysis which shows that TDR or TAR increases as $\beta_i$ increases. Our result also implies that HiDDeN watermarking may be unable to accurately encode/decode very long watermarks.

**Impact of detection threshold $\tau$:** Figure 6c shows the average TDR, average TAR, worst 1% TDR, worst 1% TAR, and FDR when the detection threshold $\tau$ varies from 0.7 to 0.95. When $\tau$ increases, both TDR and TAR decrease, while FDR also decreases. Such trade-off of $\tau$ is consistent with Theorem 1, 3, and 4.

## M  COMMON POST-PROCESSING AND ADVERSARIAL TRAINING

**Common post-processing:** Each of these post-processing methods has some parameters, which control the size of perturbation added to a (watermarked or unwatermarked) image.

**JPEG.** JPEG method (Zhang et al., 2020) compresses an image via a discrete cosine transform. The perturbation introduced to an image is determined by the *quality factor $Q$*. An image is perturbed more when $Q$ is smaller.

Table 2: Default parameter settings for the training of AWT.

| Phase | Standard Training | Fine-Tuning |
|---|---|---|
| Optimizer | Adam | |
| # epochs | 200 | 10 |
| Batch size | 16 | |
| Learning rate | $3 \times 10^{-5}$ | |
| # warm-up iterations | 6000 | 1000 |
| Length of text | 250 | $250 \pm 16$ |
| Generation weight | 1.5 | 1 |
| Message weight | 10000 | |
| Reconstruction weight | 1.5 | 2 |

**Gaussian noise.** This method perturbs an image via adding a random Gaussian noise to each pixel. In our experiments, the mean of the Gaussian distribution is 0. The perturbation introduced to an image is determined by the parameter *standard deviation $\sigma$*.

**Gaussian blur.** This method blurs an image via a Gaussian function. In our experiments, we fix kernel size $s = 5$. The perturbation introduced to an image is determined by the parameter *standard deviation $\sigma$*.

**Brightness/Contrast.** This method perturbs an image via adjusting the brightness and contrast. Formally, the method has contrast parameter $a$ and brightness parameter $b$, where each pixel $x$ is converted to $ax + b$. In our experiments, we fix $b = 0.2$ and vary $a$ to control the perturbation.

**Adversarial training (Zhu et al., 2018):** We use adversarial training to train HiDDeN. Specifically, during training, we randomly sample a post-processing method from no post-processing and common post-processing with a random parameter to post-process each watermarked image in a mini-batch. Following previous work (Zhu et al., 2018), we consider the following range of parameters during adversarial training: $Q \in [10, 99]$ for JPEG, $\sigma \in [0, 0.5]$ for Gaussian noise, $\sigma \in [0, 1.5]$ for Gaussian blur, and $a \in [1, 20]$ for Brightness/Contrast.

## N  DETECTION AND ATTRIBUTION OF AI-GENERATED TEXTS

Our method can also be used for the detection and attribution of AI-generated texts based on text watermarking that uses bitstring as watermark. For text watermarking, we use a learning-based method called *Adversarial Watermarking Transformer (AWT)* (Abdelnabi & Fritz, 2021). Given a text, AWT encoder embeds a bitstring watermark into it; and given a (watermarked or unwatermarked) text, AWT decoder decodes a watermark from it. Following the original paper (Abdelnabi & Fritz, 2021), we train AWT on the word-level WikiText-2 dataset, which is derived from Wikipedia articles (Merity et al., 2017). We use most of the hyperparameter settings in the publicly available code of AWT except the weight of the watermark decoding loss. To optimize watermark decoding accuracy, we increase this weight during training. The detailed hyperparameter settings for training can be found in Table 2.

We use A-BSTA to select users' watermarks. For each user, we sample 10 text segments from the test corpus uniformly at random, and perform watermark-based detection and attribution. Moreover, we use the unwatermarked test corpus to calculate FDR. Figure 7 shows the detection and attribution results when there is no post-processing and *paraphrasing* (Damodaran, 2021) is applied to texts, where $n = 64$, $\tau = 0.85$, and $s$ ranges from 10 to 100,000. Due to the fixed-length nature of AWT's input, we constrain the output length of the paraphraser to a certain range. When paraphrasing is used, we extend adversarial training to train AWT. Specifically, we employ T5-based paraphraser to post-process the watermarked texts generated by AWT. Due to the non-differentiable nature of the paraphrasing process, we cannot jointly adversarially train the encoder and decoder since the gradients cannot back-propagate to the encoder. To address the challenge, we first use the standard training to train AWT encoder and decoder. Then, we use the encoder to generate watermarked texts, paraphrase them, and use the paraphrased watermarked texts to fine-tune the decoder. The detail parameter settings of fine-tuning are shown in Table 2.

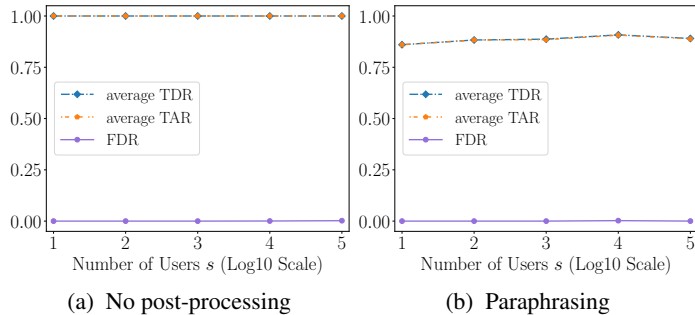

Figure 7: Results of watermark-based detection and attribution for AI-generated texts.

Note that the average TDR/TAR and FDR are all nearly 0 when AWT is trained by standard training and paraphrasing is applied to texts. The results show that our method is also applicable for AI-generated texts, and adversarially trained AWT has better robustness to paraphrasing.

## O   DISCUSSION AND LIMITATIONS

**Attribution of GenAI services:**  In this work, we focus on attribution of content to users for a specific GenAI service. Another relevant attribution problem is to trace back the GenAI service (e.g., Google's Imagen, OpenAI's DALL-E 3, or Stable Diffusion) that generated the given content. Our method can also be applied to such GenAI-service-attribution problem by assigning a different watermark to each GenAI service. When GenAI service generates content, its corresponding watermark is embedded into it. Then, our method can be applied to detect whether the content is AI-generated and further attribute the GenAI service if the content is detected as AI-generated.

In service attribution, selecting watermarks for different GenAI services can be coordinated by a central authority, who runs our watermark selection algorithm to pick unique watermarks for the GenAI services. A GenAI service registers to the central authority in order to obtain a unique watermark. Such a central authority is similar to the certificate authority in the Public Key Infrastructure (PKI) that is widely used to secure communications on the Internet. The central authority may also perform detection and attribution of AI-generated content since it has access to all GenAI services' watermarks. However, the central authority may become a bottleneck in such detection and attribution. To mitigate this issue, the watermarks of all GenAI services can be shared with each GenAI service, so each GenAI service can perform detection and attribution. We note that a central authority is not needed in user attribution of a particular GenAI service. This is because the GenAI service can select watermarks for its users and perform detection/attribution.

**Hierarchical attribution:**  We can perform attribution to GenAI service and user simultaneously. Specifically, we can divide the watermark space into multiple subspaces; and each GenAI service uses a subspace of watermarks and assigns watermarks in its subspace to its users. In this way, we can trace back both the GenAI service and its user that generated the given content.

**Developing robust watermarks:**  We acknowledge that existing watermarking methods are not robust against adversarial post-processing in white-box settings and in black-box settings where an attacker can perform many queries. However, the development of robust watermarks—though orthogonal to our work—is an active area of research and is showing significant progress. For instance, recent image watermarking techniques are certifiably robust against bounded perturbations (Jiang et al., 2024).

**C2PA (C2P, 2025):**  Beyond watermarking, C2PA creates digital signatures for content provenance. However, it lacks robustness to even minimal modifications; for example, altering a single pixel invalidates the digital signature. Consequently, C2PA is primarily applicable in scenarios where the goal is to prevent adversarial forgery of content provenance.

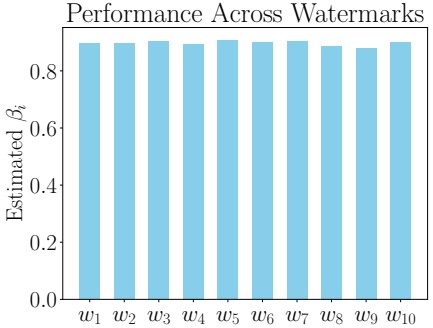

Figure 8: $w_1$ to $w_{10}$ are 10 different random watermarks and $\beta_i$ is estimated using 100 images. Different watermarks have slightly disparate performance.

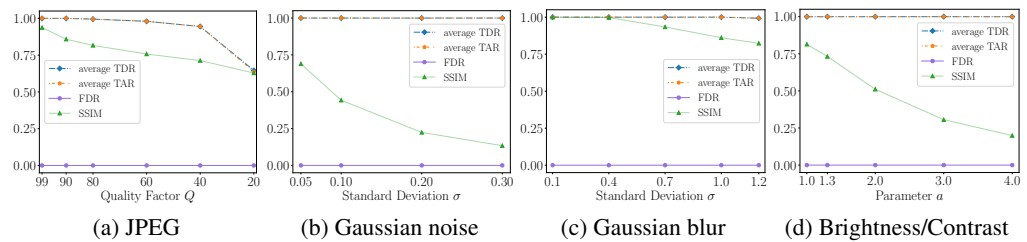

(a) JPEG      (b) Gaussian noise      (c) Gaussian blur      (d) Brightness/Contrast

Figure 9: Detection and attribution results when AI-generated and non-AI-generated images are post-processed by common post-processing methods with different parameters. SSIM measures the quality of an image after post-processing.

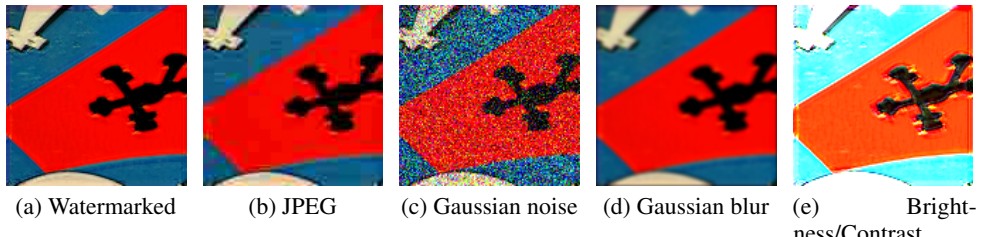

(a) Watermarked      (b) JPEG      (c) Gaussian noise      (d) Gaussian blur      (e) Brightness/Contrast

Figure 10: A watermarked image and the versions post-processed by JPEG with $Q$=20, Gaussian noise with $\sigma$=0.3, Gaussian blur with $\sigma$=1.2, and Brightness/Contrast with $a$=4.0.

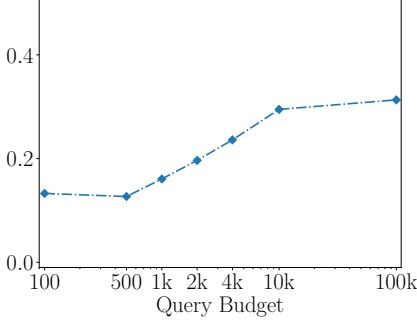

Figure 11: Average SSIM between watermarked images and their adversarially post-processed versions as query budget varies in the black-box setting.

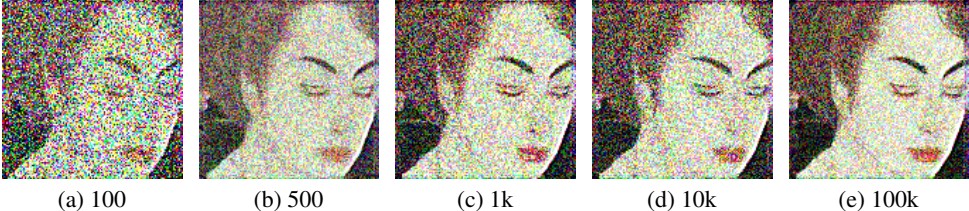

| (a) 100 | (b) 500 | (c) 1k | (d) 10k | (e) 100k |

Figure 12: Perturbed watermarked images obtained by adversarial post-processing with different number of queries to the detection API in the black-box setting.

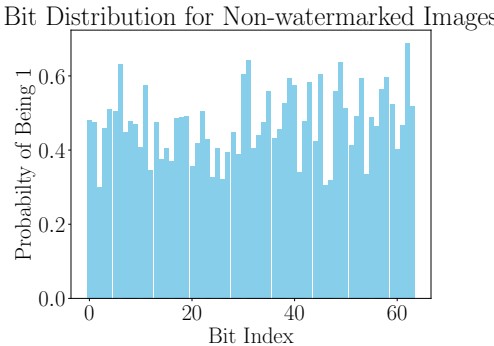

Figure 13: Bit distribution for watermarks decoded from 1,000 non-watermarked images with a watermark length of $n = 64$.

Table 3: The maximum pairwise bitwise accuracy among the watermarks generated by NRG and A-BSTA for different initializations.

|        | $\neg w_1$ initialization | Random initialization |
|--------|---------------------------|-----------------------|
| NRG    | 0.766                     | 0.750                 |
| A-BSTA | 0.875                     | 0.734                 |

Table 4: The average running time for different watermark selection methods to generate a watermark.

|           | Random | NRG  | A-BSTA |
|-----------|--------|------|--------|
| Time (ms) | 0.01   | 2.11 | 24.00  |

Table 5: Theoretical lower bounds of TDR/TAR and upper bound of FDR when there are 100 million users.

| Bound of TDR | Bound of FDR | Bound of TAR |
|--------------|--------------|--------------|
| 99.99%       | 6.00%        | 99.99%       |

Table 6: Watermark forgery results using Steganalysis. Our testing set consists of 1,000 non-watermarked images from MS-COCO dataset. We define Forgery Success Rate as the fraction of images that are detected as watermarked after forgery attack.

| Method          | Average BA↑ | Forgery Success Rate↑ | PSNR↑ | FID↓  |
|-----------------|-------------|-----------------------|-------|-------|
| Stable Signatur | 0.464       | 0.000                 | 30.65 | 2.812 |
| WAM             | 0.484       | 0.000                 | 35.01 | 2.495 |
| PRC             | –           | 0.000                 | 31.68 | 2.384 |

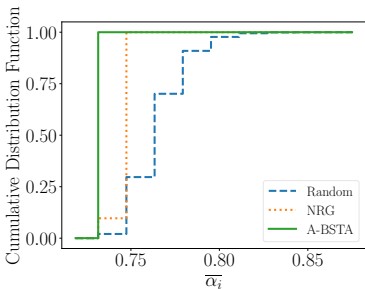

Figure 14: The cumulative distribution function (CDF) of $\overline{\alpha_i}$.

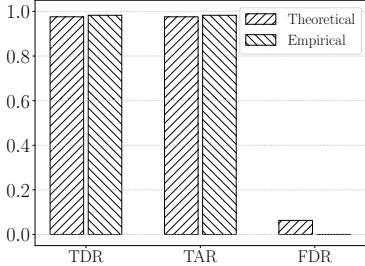

Figure 15: Theoretical vs. empirical results when JPEG with $Q = 90$ is applied.

Table 7: Performance across different watermark selection methods. The numbers show $\overline{\alpha_i} = \max_{j \in \{1, 2, \cdots, s\}/\{i\}} BA(w_i, w_j)$.

| Number of generated watermarks | 10 | 100 | 1000 | 10000 | 100000 |
|---|---|---|---|---|---|
| Random | 0.656 | 0.750 | 0.766 | 0.828 | 0.875 |
| Hash | 0.609 | 0.719 | 0.813 | 0.828 | 0.875 |
| A-BSTA | 0.531 | 0.609 | 0.672 | 0.703 | 0.734 |

