# OpenReview forum: "Watermark-based Attribution of AI-Generated Content"
_ICLR.cc/2026/Conference — ICLR 2026 Poster_

### Official Review · Reviewer_Yi1D · 2025-10-21

**Soundness:** 3
**Presentation:** 4
**Contribution:** 3
**Rating:** 6
**Confidence:** 4

**Summary:**

The paper presents a watermark attribution method for AI-generated images. In addition to the detection of watermark, it is able to attribute which user a watermarked image is generated from. In terms of evaluation, both detection rate (of whether or not a watermark has been injected) and attribution rate (of whether or not the correct user has been identified). Theoretical development is also provided.

**Strengths:**

The paper presents a cool idea, and it is well written. The proposed idea seems to be easily implemented. Theoretical results are provided to support the numerical findings.

**Weaknesses:**

The definition of tau is not intuitive. How to select tau seems to be a tricky question -- an absolutely large tau may rule out the possibility of detecting the watermark, whereas a small tau may render multiple i's satisfying the TDR condition, making the selection of i as in the TAR stage sensitive to ties and run-ups. This becomes more challenging when the dimension of watermark is long (e.g., 64-bit, as mentioned by the authors). The results seem to show that when tau increases, both TDR and TAR decrease while FDR also decreases. This is not intuitive. It would be helpful if additional numerical experiments can be conducted to evaluate the robustness against the selection of tau in real data.)

As for post-processing, the authors consider limited non-adversarial settings. It would be helpful to also consider the robustness of the proposed procedure against flipping, rotation, and cropping.

The work claims to watermark and attribute AI-generated images. However, it is not clear to me where and how the proposed method takes advantage of the AI image-generating process. As demonstrated in the numerical experiments, the proposed method seems to work well even if the datasets are not AI generated.

**Questions:**

Is the proposed method plug-and-play? In other words, does it have to be tied to certain image-generating processes, such as diffusion? If I understand correctly, the answer seems to be no. But please clarify.

It is not clear to me how the binary watermarks are injected to the encoded images. Would a non-binary watermark work as well?

Any requirements on the number of images generated by each user? In other words, if we certainly know that multiple images are from the same user (but don't know who the user is), would the proposed method be able to incorporate such information and hence improve the TAR?

---

> ### Author Response · Authors · 2025-11-20
>
> Thank you for your thoughtful feedback and comments. We are encouraged that you recognized our work as proposing a practical and interesting idea, presented with high writing quality and supported by solid theoretical analysis. Your comments have been invaluable in helping us strengthen the paper, and we have addressed them as outlined below. Please feel free to share any additional concerns—we would be more than willing to discuss them further. We also kindly ask you to consider revisiting the rating in light of these improvements.
>
> ---
>
> Comment#1: The results of $\tau$ are not intuitive.
>
> Answer#1: We will clarify these points in the revised version. Note that True Detection Rate (TDR) and False Detection Rate (FDR) only concern the detection stage, whereas True Attribution Rate (TAR) involves the attribution stage as well. When $\tau$ increases, both watermarked and non-watermarked images become less likely to be detected as watermarked, causing TDR and FDR to decrease. Regarding TAR, you are partially correct that “a small $\tau$ may allow multiple (i) to satisfy the TDR condition, making the selection of (i) in the TAR stage sensitive to ties and run-ups.” However, in our framework, attribution is determined by identifying which watermark is most similar to the decoded watermark. With our definition, it is highly unlikely to encounter ties when $\tau>\frac{1+\overline{\alpha_i}}{2}$, as formalized in Theorem 4:
>
> $$
> TAR_i \geq \operatorname{Pr} ( n_i \geq \max { \left\lfloor \frac{1+\overline{\alpha_i}}{2} n \right\rfloor+1, \tau n } )
> $$
>
> We call this interesting observation “detection implies attribution” (explained in line 755-760 in the Appendix). Specifically, this indicates that once an AI-generated image is correctly detected, it would also be correctly attributed. Also, since we only perform attribution after an image has been detected as watermarked, TDR$_i$ and TAR$_i$ are very close (validated by our experiments) when $\tau$ is relatively large (0.9 in our experiments).
>
> Impact of detection threshold $\tau$ on real data is shown in Figure 6 in Appendix L. When $\tau$ increases, TDR, FDR, and TAR all decrease. Such observations actually align with Theorem 1,3,4 and theoretical analysis.
>
> ---
>
> Comment#2: Robustness against flipping, rotation, and cropping.
>
> Answer#2: Our framework inherits both the accuracy and (non-)robustness properties of the underlying watermark. We emphasize that our work leverages existing watermarking methods to achieve user-level attribution of AI-generated images. Developing new watermarking techniques that are robust to post-processing, i.e., capable of extracting watermarks even after the image has been modified, is orthogonal to our current focus. Nevertheless, we are happy to include such experiments in the revised version to address your concern. For example, robustness results for HiDDeN are as follows:
>
> | Flipping | Horizontal | Vertical |
> |-------------|---------|---------|
> | Avg Bitwise Accuracy | 0.999 | 0.999 |
> | Avg TDR | 0.999 | 0.999 |
> | Avg TAR | 0.999 | 0.999 |
>
> | Rotation | 0 | 90 | 180 | 270 |
> |-------------|---------|---------|---------|---------|
> | Avg Bitwise Accuracy | 0.999 | 0.996 | 0.995 | 0.989 |
> | Avg TDR | 0.999 | 0.999 | 0.999 | 0.998 |
> | Avg TAR | 0.999 | 0.999 | 0.999 | 0.998 |
>
> | Cropping | 1.00 | 0.99 | 0.98 | 0.95 | 0.9 |
> |-------------|---------|---------|---------|---------|---------|
> | Avg Bitwise Accuracy | 0.999 | 0.995 | 0.992 | 0.984 | 0.950 |
> | Avg TDR | 0.999 | 0.999 | 0.998 | 0.990 | 0.959 |
> | Avg TAR | 0.999 | 0.999 | 0.998 | 0.989 | 0.956 |
>
> The results show that our framework inherits the robustness properties of the underlying HiDDeN watermark against flipping, rotation, and cropping post-processing.
>
> ---
>
> Comment#3: How is the image generation process involved?
>
> Answer#3: There are two main categories of image watermarking methods: post-generation and in-generation. Post-generation watermarking methods, such as HiDDeN and StegaStamp (both evaluated in our work), can embed watermarks into any given image, regardless of whether it is AI-generated. Thus, a GenAI service can first generate an image based on a user’s prompt and then apply the watermarking module to embed the watermark before returning the image to the user. In contrast, in-generation watermarking methods, such as PRC (also evaluated in our work), embed the watermark directly during the image generation process. As a result, all images produced by the model are already watermarked, but these methods cannot be applied to pre-existing images. Importantly, our framework is compatible with both categories of watermarking methods, enabling broad applicability across different GenAI service designs.
>
> ---
>
> Comment#4: Is the framework plug-and-play?
>
> Answer#4: Yes, our framework is applicable and plug-and-play to all image watermarking methods that use bitstrings as watermarks. It is not tied to any specific image generation process.

---

> > ### Comment · Reviewer_Yi1D · 2025-11-25
> >
> > I thank the authors for their efforts on addressing my comments. Most of my concerns are addressed. There are only two points that may need further clarifications.
> >
> > 1. The explanation on tau is still confusing, even after I read the related paragraphs and figures the authors pointed me to. Specifically, it is still not clear to me where the tradeoff is and how it is manifested. In Figure 6, it looks like one should choose tau as large as possible, since TDR and TAR are minimally impacted, whereas FDR is (almost) reduced to zero.
> >
> > 2. For the binary watermarks, could you please explain how "the watermark encoder embeds this bitstring into a given image to produce an encoded image?"

---

> ### Author Response · Authors · 2025-11-20
>
> Comment#5: Binary and non-binary watermarks.
>
> Answer#5: We treat a binary watermark as a bitstring. Concretely, the watermark encoder embeds this bitstring into a given image to produce an encoded image, while the watermark decoder aims to recover the bitstring from it. The encoder–decoder pair is jointly trained so that the decoded bitstring closely matches the ground-truth bitstring. Although our experiments focus on binary (bitstring-based) watermarks, which are the mainstream, the framework is not limited to them—non-binary watermarks can also be supported since they can be turned into binaries.
>
> ---
>
> Comment#6: Influence of the number of images.
>
> Answer#6: That is a very interesting point. If multiple images are known to originate from the same user, they can indeed improve attribution accuracy. For example, one can perform attribution on each image independently and then aggregate the results (e.g., via majority vote). For example, using the HiDDeN watermark on the Stable Diffusion dataset with parameters $s=100,000$, $n=64$, and $\tau=0.9$, our average TAR increases from 0.998 to 1.000 when majority vote aggregation is applied. We have added a discussion of this practical scenario in the revised version.

---

> ### Author Response · Authors · 2025-11-25
>
> Thank you for the follow-up comments. We are glad to hear that we have addressed most of your concerns, and we now have a clearer understanding of your comment regarding the intuition behind the results for $\tau$.
>
> ---
>
> 1. In our theorems and experiments, we show that when $\tau$ increases, TDR and TAR both decrease, and FDR also decreases. You found this not intuitive and mentioned that “a small $\tau$ may render multiple $i$’s satisfying the TDR condition, making the selection of $i$ as in the TAR stage sensitive to ties and run-ups.” We think you are probably suggesting that TAR may increase when $\tau$ increases, and we believe there might be a misunderstanding.
>
> You are correct that a small $\tau$ may allow multiple $i$’s to satisfy the TDR condition. However, this does not make the selection of $i$ in the TAR stage difficult or sensitive to ties. This is because our attribution depends solely on identifying which user-specific watermark is the **most** similar to the decoded watermark; this ranking is fixed for a given decoded watermark and does not depend on how many $i$ satisfy the TDR condition.
>
> To illustrate, given a decoded watermark, its similarity to each user’s watermark is fixed, and thus the index $i$ whose watermark is the most similar is also fixed. Changing $\tau$ only affects how many $i$ pass the TDR threshold—it does not alter the ranking of similarities and therefore does not affect the attribution result. As long as at least one $i$ satisfies the TDR condition, the attribution outcome remains the same, no matter what value of $\tau$ is picked.
>
> The reason we state that “when $\tau$ increases, TAR decreases” is that increasing $\tau$ lowers TDR; if a watermark is not detected at all, attribution cannot be performed, which consequently reduces TAR.
>
> We hope this addresses the confusion. Please feel free to share any additional concerns—we would be more than willing to discuss them further.
>
> ---
>
> 2. In our work, Both HiDDeN[1] and StegaStamp[2] use a learned neural watermark encoder to embed a bitstring into an image. The encoder $E$ takes the input image $I$ and the bitstring $w$ as input, generating an image-size small perturbation $\delta$:
>
> $\delta = E(I, w)$.
>
> This perturbation is then added to the original image to produce the watermarked image $I_w$:
>
> $I_w = I + \delta$.
>
> During training, the encoder is optimized so that (1) the perturbation is imperceptible, and (2) a decoder can reliably recover the bitstring. Thus, “embedding the bitstring” simply refers to learning pixel-level modifications that encode the bitstring in a way that is visually invisible but decodable and robust.
>
> PRC Watermark[3] is an in-processing watermark: it does not post-process an existing image, but embeds the bitstring during image generation, by modifying the initial latent fed into a diffusion model.
>
> Before generation, the method chooses the initial noise latent vector using a pseudorandom error-correcting code (PRC) rather than sampling from a standard Gaussian. That is, the latent is not random in the usual sense — it's deterministically generated (given a secret key + bitstring) so that its structure encodes a hidden “bitstring watermark.” That pseudorandom latent seed ensures that the generated image will carry a watermark. For decoding/detection, one inverts the generation (e.g., using the same diffusion model) to recover an approximation of the initial latent (or at least its relevant bits), then applies the PRC decoding algorithm (with the secret key) to recover the embedded bitstring.
>
> The design of image watermarks is not the focus of this work. Our framework is general and plug-and-play, and can be applied to all those image watermarking methods.
>
> ---
>
> [1] Jiren Zhu, Russell Kaplan, Justin Johnson, and Li Fei-Fei. Hidden: Hiding data with deep networks. ECCV, 2018.
>
> [2] Matthew Tancik, Ben Mildenhall, and Ren Ng. Stegastamp: Invisible hyperlinks in physical photographs. CVPR, 2020.
>
> [3] Sam Gunn, Xuandong Zhao, and Dawn Song. An undetectable watermark for generative image models. ICLR, 2025.

---

### Official Review · Reviewer_p6Ev · 2025-11-02

**Soundness:** 1
**Presentation:** 2
**Contribution:** 1
**Rating:** 2
**Confidence:** 4

**Summary:**

The paper proposes a user-oriented watermark assignment approach such that the images a user generates are assigned to the image and is then attributed to the user when needed. The approach is a good one.

**Strengths:**

Pros: In practical scenarios, provenance of data and its attribution can be assigned to a user.

**Weaknesses:**

Cons:
1. The paper does not discuss privacy at all. Such attribution discloses privacy and thus other sensitive information. The watermarks can lead to replay attacks - malicious users can attribute the watermark of one user to an image that they did not generate.
2. The paper does not look into simple mechanisms - such as hashing schemes - why such hashes of enough size not be used as watermarks? There have been work in security community. So the watermark selection problem for scalability across users would not be a problem. That is a straightforward solution, unless the authors convince the readers why it is not a problem.
3. If there is a watermark can be extracted, or should not be extracted but hidden - there is a body of work in steganography.

**Questions:**

Look at the cons.

---

> ### Author Response · Authors · 2025-11-20
>
> Thank you for your thoughtful feedback and comments. Your comments have been invaluable in helping us strengthen the paper, and we have addressed them as outlined below. Please feel free to share any additional concerns—we would be more than willing to discuss them further. We also kindly ask you to consider revisiting the rating in light of these improvements.
>
> ---
>
> Comment#1: Privacy issues.
>
> Answer#1: We appreciate the reviewer’s concern. Attribution systems inherently involve a trade-off between provenance and user privacy, since provenance requires tracing content back to a specific user, which naturally raises privacy concerns. Importantly, our watermark-based attribution does not require exposing user identity or sensitive information to arbitrary third parties. The attribution metadata can be designed so that only trusted entities (e.g., the GenAI service provider or a designated verification authority, similar in role to the Public Key Infrastructure) are able to perform verification. This follows established practices in responsible provenance systems. We have added this discussion in Discussion in the revised version.
>
> ---
>
> Comment#2: Practicality of replay attacks.
>
> Answer#2: If we understand correctly, the replay attack you mentioned corresponds to a watermark forgery attack; that is, a malicious user attempts to forge the watermark of a specific user for a given image based on existing watermarked images produced by that user, such that the forged image would be falsely attributed to them even though it was not actually generated by that user. According to state-of-the-art research on watermark forgery attacks, forgery in white-box settings, where the attacker has full access to the watermarking system, is easy but not very practical. However, in non-white-box settings, where the attacker has only limited knowledge of the watermarking system, forgery remains challenging even at the detection level, let alone for user-level attribution, since the malicious user does not know the specific user’s watermark. For example, the state-of-the-art watermark forgery attack Steganalysis [1] (NeurIPS 2024) reports effectiveness against several content-agnostic watermarking methods, such as TreeRing [2], GaussianShading [3], and RingID [4]. However, it fails against many content-dependent state-of-the-art image watermarking methods, as confirmed by our experiments:
>
> |             | AvgAcc↑ | Forgery Success Rate↑ | PSNR↑ | FID↓  |
> |-------------|---------|--------------|-------|-------|
> | Stable Signature[5] | 0.464 | 0.000 | 30.65 | 2.812 |
> | WAM[6] | 0.484 | 0.000 | 35.01 | 2.495 |
> | PRC[7] | - | 0.000 | 31.68 | 2.384 |
>
> The results indicate that even state-of-the-art image watermark forgery attacks are unable to successfully forge the content-dependent watermarks (forgery success rate = 0). Thus, the replay attack does not appear to pose a significant concern. We regard this as an interesting direction for future work and have included a discussion of it in Discussion in the revised version.
>
> ---
>
> [1] Pei Yang, Hai Ci, Yiren Song, and Mike Zheng Shou. Can simple averaging defeat modern watermarks? NeurIPS, 2024.
>
> [2] Yuxin Wen, John Kirchenbauer, Jonas Geiping, and Tom Goldstein. Tree-rings watermarks: Invisible fingerprints for diffusion images. NeurIPS, 2023.
>
> [3] Zijin Yang, Kai Zeng, Kejiang Chen, Han Fang, Weiming Zhang, and Nenghai Yu. Gaussian shading: Provable performance-lossless image watermarking for diffusion models. CVPR, 2024.
>
> [4] Hai Ci, Pei Yang, Yiren Song and Mike Zheng Shou. RingID: Rethinking Tree-Ring Watermarking for Enhanced Multi-key Identification, ECCV, 2024.
>
> [5] Pierre Fernandez, Guillaume Couairon, Herve J´egou, Matthijs Douze, and Teddy Furon. The stable signature: Rooting watermarks in latent diffusion models. ICCV, 2023.
>
> [6] Tom Sander, Pierre Fernandez, Alain Oliviero Durmus, Teddy Furon, and Matthijs Douze. Watermark anything with localized messages. ICLR, 2025.
>
> [7] Sam Gunn, Xuandong Zhao, and Dawn Song. An undetectable watermark for generative image models. ICLR, 2025.

---

> ### Author Response · Authors · 2025-11-20
>
> Comment#3: Hashing schemes.
>
> Answer#3: If we understand correctly, the reviewer is suggesting using a hash function to generate user-specific watermarks (given the user ID). However, such a simple mechanism essentially picks watermarks for users randomly due to the pseudo randomness of hash functions. Therefore, this mechanism behaves similarly to our Random baseline (the details for Random and other baselines are in Appendix K), which generates a watermark (bitstring) uniformly at random for each user. Thus, both hash and Random baseline should only achieve sub-optimal attribution performance. Their problem is that two users may still receive highly similar watermarks. According to Theorem 4, to maximize the lower bound of TAR, we need to minimize $\overline{\alpha_i}=\max _{j \in\{1,2, \cdots, s\} /\{i\}} B A\left(w_i, w_j\right)$ the maximum similarity between each user’s watermark and all others. Our A-BSTA algorithm is specifically designed for this purpose, and our experimental results in Figure 3(a) validate that A-BSTA improves attribution performance. To further illustrate this, we also conducted experiments using hash-based watermarks (64 bits):
>
> | Number of generated watermarks | 10 | 100 | 1000 | 10000 | 100000 |
> |-------------|---------|--------------|-------|-------|-------|
> | Random | 0.656 | 0.750 | 0.766 | 0.828 | 0.875 |
> | Hash | 0.609 | 0.719 | 0.813 | 0.828 | 0.875 |
> | A-BSTA | 0.531 | 0.609 | 0.672 | 0.703 | 0.734 |
>
> The numbers indicate $\overline{\alpha_i}$↓, showing that our watermark selection method A-BSTA significantly outperforms both the Random baseline and Hash. We have included those results in the revised version.
>
> ---
>
> Comment#4: Works in steganography.
>
> Answer#4: We thank the reviewer for pointing this out. Steganography and image watermarking both embed information into images, but they are designed for different goals. Steganography prioritizes secrecy and undetectability, whereas watermarking prioritizes robustness under post-processing. As a result, most steganographic schemes are not designed to withstand standard image transforms (e.g., resizing, compression, filtering), which makes them unsuitable for attribution in real-world GenAI pipelines. In contrast, watermarking methods— including the one we build upon— are explicitly designed for robustness and can survive such transformations, as demonstrated by our experiments.
>
> In Discussion part of the revised version, we have added a discussion clarifying this conceptual distinction and why robust watermarking, rather than steganography, is appropriate for AI-generated image provenance and attribution. If there are representative works in steganography that you would like us to discuss, please feel free to let us know; we would be happy to address them.

---

### Official Review · Reviewer_hwBd · 2025-11-04

**Soundness:** 3
**Presentation:** 3
**Contribution:** 2
**Rating:** 6
**Confidence:** 3

**Summary:**

This paper looks at attribution of ai-generated image. The paper proposes to use a personalized watermark for each user when generating images with AI. After that, the paper theoretically derives lower bounds on watermark detection and user attribution performance, and select watermarks for users to maximize these lower bounds. Empirical analysis shows that the proposed method inherits both the accuracy and (non-)robustness properties of the underlying watermark.

**Strengths:**

- The paper looks at watermark detection and specifically attribution to the users, which is under explored in the literature.
- The paper provides theoretical analysis on the upper and lower bounds on the detection rate.
- Empirical results seem to suggest that the proposed method preserves utility and robustness.

**Weaknesses:**

- The motivation of attribution of ai-generated images to user is unclear. I can understand why detection of ai-generated images is important and that is a main motivation of inserting watermark for ai generated content. But it's unclear to me why attribution to users is an important, practical concern. Some discussion on this would be important.
- Section 7.1, comparison with non user specific watermark is missing. It would be interesting to study whether such robustness to post-processing is preserved / enhanced / weakened by using a user specific watermark compared to a general watermark.

**Questions:**

N/A

---

> ### Author Response · Authors · 2025-11-20
>
> Thank you for your thoughtful feedback and comments. We are encouraged that you recognized our work as addressing an under-explored problem, providing theoretical analysis, and demonstrating strong utility and robustness. Your comments have been invaluable in helping us strengthen the paper, and we have addressed them as outlined below. Please feel free to share any additional concerns—we would be more than willing to discuss them further. We also kindly ask you to consider revisiting the rating in light of these improvements.
>
> ---
>
> Comment#1: Why is attribution important?
>
> Answer#1: We have clarified and added more discussion in the Introduction in the revised version.
>
> As we stated in the Introduction and Problem Formulation, attribution aims to trace the origin of an AI-generated image by identifying the specific user account of the GenAI service that produced it. While detection is important for recognizing that an image is synthetic, attribution addresses a different and increasingly practical need.
>
> In real-world deployments, GenAI services face substantial misuse risks: malicious users can generate harmful images, e.g., political deepfakes, deceptive propaganda, or illegal content, and then distribute them anonymously. In such cases, detection alone is insufficient: knowing that an image is AI-generated does not reveal who created it. Attribution provides service providers and, where appropriate, law-enforcement agencies with actionable forensic evidence that links harmful content back to the originating user account or API key. This is essential for:
>
> - **Identifying responsible users** during cybercrime or abuse investigations, including coordinated disinformation operations.
> - **Enabling platform accountability**, such as suspending or banning abusive accounts and preventing repeated misuse.
> - **Maintaining auditability and compliance**, especially as emerging regulations (e.g., AI Act–style policies) require platforms to ensure traceability of generated content.
>
> Thus, attribution complements detection: detection determines what an image is, while attribution enables stakeholders to understand who created it. Both are necessary for building a practical, enforceable, and responsible ecosystem around generative AI services.
>
> ---
>
> Comment#2: Comparison with non user specific (user-agnostic) watermark.
>
> Answer#2: That is a good point! We actually have this comparison in line 283-285 in the main text: “Due to page limits, we systematically analyze the impact of s on detection bounds, the difference between user-agnostic and user-aware detection, and the relationship between detection and attribution in Appendix D.”
>
> In the Appendix, we theoretically and empirically compare user-specific and user-agnostic detection as follows: For our user-specific detection, the first term of the lower bound in our Theorem 1 is a lower bound of TDR for user-agnostic watermark; the first term of the upper bound in our Theorem 2 is an upper bound of FDR for user-agnostic detection; and the upper bound with $s=1$ in our Theorem 3 is an alternative upper bound of FDR for user-agnostic detection.  Therefore, compared to user-agnostic detection, our user-specific detection achieves larger TDR but also larger FDR. Figure 5 in Appendix D illustrates empirical TDR and FDR results for user-agnostic and user-specific detection. Compared with Figure 6 in the Appendix, the results indicate that robustness to post-processing is well-preserved, consistent with our theoretical analysis.

---

### Meta-Review · Area_Chair_kTqQ · 2025-12-28

**Summary:**

The paper studies the attribution of AI-generated image using watermark based method --- identifying the original user who created an image using AI generation service. The paper establishes mathematical definition of true/false detection rate and true/false attribution rate, and derives the bounds among them. Their key finding is that detection leads to attribution. The paper then formulates and proposes a method for the watermark key construction problem for s users to minimize the confusion (false attribution). The paper presents a reasonable experimental result showing the effectiveness of their attribution on top of three existing watermark methods.

Strengths of the paper:
1. The paper provides a reasonable theoretical analysis of the true/false attribution rate.
2. The experimental results show the effectiveness of the method.

Weakness of the paper:
1. The motivation of attribution is not clear. (This is addressed by authors in the revision).
2. The robustness of the method on image flipping/cropping/rotation is not shown. (the authors provided additional experiments)
3. Results comparing with the hashing-based approach and nspecific watermarking. (Authors have responded and addressed well).
4. Explanation about some technical terms (e.g. tau). (Authors have provided additional explanantion).

Reviewer p6Ev raised the following concerns:
1. privacy of user is not protected.
2. scenarios where watermark should not be extracted.
These concerns focus on the application scenario but these are not the original problem of this paper. They are orthogonal to the problem solved in this paper (which is attribution problem). The paper did not make any claim about these either. So these concerns should not be considered when assessing the technical contribution and suitability of the paper in ICLR. They should be considered in separate papers. Since Reviewer p6Ev's other concern is addressed, the overall score is down-weighed in the final decision.

Overall, I find that all other reviewers are positive about the paper, and the authors addressed all concerns well.

**Reviewer Concerns:**

All concerns are addressed, except Reviewer p6Ev's two concerns. But these two concerns are not technical --- they are based on reviewer's proposed orthogonal problem which is not studied in the paper.

**Reviewer Scores:**

Reviewer p6Ev would not change the score but the review is not objective assessment on the paper. The reviewer is giving a low quality and irrelevant review.

---

### Decision · Program_Chairs · 2026-01-26

Accept (Poster)